**Data Availability Statement:** All relevant computer program files are available from https://sourceforge.net/projects/arcgis-nra-tool/. A list of species is provided with the Supplemental

# A GIS-based policy support tool to determine national responsibilities and priorities for biodiversity conservation

Yu-Pin Lin[1⊙], Dirk S. Schmeller[ID][2⊙], Tzung-Su Ding[3‡], Yung Chieh Wang[4‡], Wan-Yu Lien[1‡], Klaus Henle[5⊙], Reinhard A. Klenke[ID][5,6⊙] *

**1** Department of Bioenvironmental Systems Engineering, National Taiwan University, Taipei, Taiwan, **2** ECOLAB, Université de Toulouse, CNRS, INPT, UPS, Toulouse, France, **3** School of Forestry and Resource Conservation, National Taiwan University, Taipei, Taiwan, **4** Department of Soil and Water Conservation, National Chung Hsing University, Taichung City, Taiwan, **5** UFZ—Helmholtz Centre for Environmental Research, Department of Conservation Biology, Leipzig, Germany, **6** German Centre for Integrative Biodiversity Research (iDiv), sDiv–The Synthesis Center of iDiv, Halle-Jena-Leipzig, Leipzig, Germany

⊙ These authors contributed equally to this work.
‡ These authors also contributed equally to this work.
* reinhard.klenke@ufz.de

## Abstract

Efficient biodiversity conservation requires that limited resources be allocated in accordance with national responsibilities and priorities. Without appropriate computational tools, the process of determining these national responsibilities and conservation priorities is time intensive when considering many species across geographic scales. Here, we have developed a computational tool as a module for the ArcGIS geographic information system. The ArcGIS National Responsibility Assessment Tool (NRA-Tool) can be used to create hierarchical lists of national responsibilities and priorities for global species conservation. Our tool will allow conservationists to prioritize conservation efforts and to focus limited resources on relevant species and regions. We showcase our tool with data on 258 bird species and various biophysical regions, including Environmental Zones in 58 Asian countries and regions. Our tool provides a decision support system for conservation policy with attractive and easily interpretable visual outputs illustrating national responsibilities and priorities for species conservation. The graphical output allows for smooth integration into assessment reports, such as the European Article 17 report, the Living Planet Index report, or similar regional and global reports.

## Introduction

State signatories to the Convention on Biological Diversity must not only define their own strategy to protect biodiversity, they must also set their own priorities considering available resources. Even more importantly, biodiversity rich countries with few resources may be able to justify the demand of additional resources for international institutions. There is still room to increase the global efficiency of biodiversity conservation by aligning national conservation

Information in combination with the manuscript. We used data described and provided from: Metzger MJ. The Global Environmental Stratification: A high-resolution bioclimate map of the world [dataset]. The University of Edinburgh; 2018. doi: 10.7488/ds/2354 Olson D, Dinerstein E, Wikramanayake E, Burgess N, Powell G, Underwood E, D'amico J, Itoua I, Strand H, Morrison J, Loucks C, Allnutt T, Ricketts T, Kura Y, Lamoreux J, Wettengel W, Hedao P, Kassem K. Terrestrial ecoregions of the world – A new map of life on Earth. Bioscience 2001;51(11):933-938. doi: 10.1641/0006-3568(2001)051[0933:TEOTWA]2.0. CO;2. The sources for ecoregions, technical descriptions, and digital data mentioned in this source are available from: https://www. worldwildlife.org/pages/conservation-science-data-and-tools IUCN (International Union for Conservation of Nature and Natural Resources). Spatial Data & Mapping Resources. Spatial Data Download. IUCN, Gland, Switzerland and Cambridge, UK; 2020a [cited 2020 November 20]. Available from: https://www.iucnredlist.org/resources/spatial-data-download IUCN (International Union for Conservation of Nature and Natural Resources). Red List database. IUCN, Gland, Switzerland and Cambridge, UK; 2020b [cited 2020 November 20]. Available from: https://www.iucnredlist.org.

**Funding:** Y-P L, T-S D, Y-Ch W, W-Y L were mainly funded by the National Science Council of Taiwan under grant NSC101-2923-I-002-001-MY2. In 2014 the NSC was renamed in Ministry of Science and Technology (MOST, http://www.most.gov.tw/?l=en) K H, D S S, R A K were funded by the project SCALES: Securing the Conservation of biodiversity across Administrative Levels and spatial, temporal, and Ecological Scales, under the European Union's Framework Program 7 (grant 226852; https://cordis.europa.eu/project/id/226852, www.scales-project.net).

**Competing interests:** The authors have declared that no competing interests exist.

actions with current knowledge of species distribution ranges and habitats. Species and habitats are not equally distributed in and across countries and continents. Some are widespread with a distribution area spanning several countries, while others have a small distribution range concentrated within a single country. The political responsibility for the conservation of any one species is rarely completely clear, which impacts the efficiency of protection measures [1]. Thus far, conservation efforts have been based on the use of red lists, a popular indicator of the threat status of a species and an easy way to define conservation priorities, especially on finer geographic scales. Red lists can facilitate simple explanations of the complex phenomenon of "endangerment" [2, 3], resulting in high public acceptance [4–6]. However, red lists are at best a suboptimal tool for setting conservation priorities in a country or region since a threat status does not always accurately reflect conservation requirements across the entire distribution range [7–12].

A complementary concept is that of national responsibilities (NRs). National responsibilities consider the size of the distribution range of a species within a country to be proportional to that country's conservation responsibilities. Hence, a country covering a large proportion of the total distribution area of a threatened species would have a greater responsibility for the protection of this species then a similarly sized country that contains a smaller proportion of the total distribution area of this species. A methodological review of the existing approaches to the determination of national responsibilities [13] has shown the drawbacks of existing methods, especially with regard to data needs, comparability across countries, and the disentanglement of red lists and national responsibilities. Generally, the concept of national responsibility as a spatial prioritization tool takes into consideration the variability of contributions to overall species or habitat viability and persistence across distribution ranges. Hence, particular parts of species or habitat ranges (i.e., areas with high abundance of a species) are more important than others for the global conservation of a species. The concept of national responsibility captures the quality of irreplaceability that defines certain parts of a distribution range [14], and serves as a proxy for the global persistence probability of species or habitats when a particular area is lost.

National responsibility assessments identify which country or countries should lead conservation actions directed at particular species and habitats. Once national responsibilities have been determined, countries can employ additional tools that utilize the available information to select species, habitats, and regions in which practical conservation actions should be prioritized (Table 1).

Hence, determining national responsibilities [36] and conservation priorities (CPs, [37]) allows nations to take action where it is most urgently needed and helps decision makers to allocate resources efficiently towards species monitoring, management, and protection [1, 38]. For example, this process could support the United Nations Environment Programme (UNEP) in the identification of biodiversity targets for developmental programs. Entities such as the Intergovernmental Science-Policy Platform for Biodiversity and Ecosystem Services (IPBES), the Group of Earth Observations Biodiversity Observation Network (GEO BON), and international non-governmental organizations could use assessments of national responsibilities and conservation priorities as a basis for evaluating which regions would reap the greatest benefit from capacity-building efforts [1, 7].

The sheer number of species, known and unknown, makes it nearly impossible to manually determine national responsibilities and conservation priorities across all countries and regions. Such a global assessment would be highly useful for directing international and domestic conservation policy, and since a manual assessment is not feasible, a tool is needed to automate the process. To be successful, the tool must be able to overlay species or habitat distributions onto biogeographic regions with political boundaries in a geographic information system

**Table 1. Approaches for the determination of conservation priorities and respective references.**

| Approach | References |
|---|---|
| Combined use of spatial distributions of species, habitats, ecosystems and their services, connectivity measures, species compositions and economic costs | [15–20] |
| Spatial characteristics of species distributions | [21, 22] |
| Community composition | [23] |
| Patterns of genetic and morphological variation | [24, 25] |
| Geographic and evolutionary rarity | [26, 27] |
| Trait or phylogenetic distinctiveness | [28] |
| Trait-based metrics | [29] |
| Movement behavior | [30] |
| Conservation condition, biodiversity value, pressure factor, and cover relevance of habitat types | [31] |
| Social and ecological dynamism | [32] |
| Human disturbances | [33] |
| Complementary comparison of flagship and background species | [34] |
| Multivariate statistics based on variables such as state of knowledge, forest loss, forest loss acceleration, protected area size and relative species diversity | [35] |

(GIS) [7, 39, 40]. Furthermore, for comparability between regions and applicability on different ecological scales or administrative levels, such an approach needs to be freely scalable to account for the size of administrative boundaries [7, 36].

Although there are highly specialized instruments for spatial prioritization, allocation of conservation efforts and spending, and reserve planning (e.g. Marxan [17] and Zonation [15]), their application at global scale is difficult and they do not provide a clear definition of the conservation responsibilities of particular regions or countries.

Here, we describe a GIS tool to determine national responsibilities and conservation priorities globally. The method underlying the GIS tool [36, 37] is limited only by the availability of distribution data for species and habitats, and is much less data demanding than other approaches, while still scientifically robust [13]. The ArcGIS National Responsibility Assessment Tool (NRA-Tool) uses freely available shapefiles of species distributions, environmental zones, and political borders to automate the analysis for a large set of species for which data are available. Data sources include the Spatial Data & Mapping Resources and the Red List database from the IUCN [41, 42].

We explain the GIS module's user interface and illustrate the general concept and functionality with data from four species of shrews, which have their distribution center either only in Europe or are widespread across Eurasia. As a case study, we also provide national responsibility assessments for 258 bird species of Asia.

## Methods

### Implementation of the national responsibility and conservation priority method

In general, our method can be used to assess the responsibility of an administrative unit on any spatial scale. The scale of analysis will be defined by the "focal area" (FA; often the country for which a responsibility assessment must be made) and the "reference area" (RA). The reference area is the region in which the focal area must be fully contained and the size to which the focal area relates. For the assessment of national responsibilities, a sub-continental (e.g. Asia, Europe, North and South America), continental (e.g. Eurasia, Australia) or global scale is appropriate. For some applications it may be useful to choose a reference area that is defined

by a common legal and economic framework and differs in shape and extent from all continents and sub-continents (e.g. the European Union). To avoid pitfalls caused by special constellations, we recommend using the global scale as a reference area in the following cases: (i) different borderlines for a subcontinent (e.g. Europe including parts of Turkey and Kazakhstan versus an extent based strictly on national borders) [43]), (ii) inclusion or exclusion of parts of a country that are on other continents (e.g., the overseas departments and territories of France), or (iii) countries divided by a sub-continental border (e.g. Russia).

Examples of the different tasks during the process of GIS analysis are shown in Figs 1–3 and Figs 5 and 6, while the identification of national responsibilities and conservation priorities implemented in the NRA tool is based on the decision tree shown in Fig 4 and the scores or classes derived from them in Table 2.

Schmeller et al. [13] reviewed different approaches for determining national responsibilities and suggested a standardized method that is freely scalable and can provide an improved basis for setting conservation priorities. The method is also applicable to habitats or ecosystems [7] and comprises three steps (Fig 4).

The expected distribution proportion ($DP_{exp}$) in the focal area is the ratio of the distribution area of a species inside the reference area (*sensu* Schmeller et al. [36, 37]) to the total area of the reference area (Eq 1); the observed distribution proportion ($DP_{obs}$) in the focal area is the ratio of the distribution area of the species in a focal area to the total area of the focal area (Eq 2) [36, 37, 47].

$$DP_{exp} = \frac{Distribution\ Area \cap Reference\ Area}{Reference\ Area} \tag{1}$$

$$DP_{obs} = \frac{Distribution\ Area \cap Focal\ Area}{Focal\ Area} \tag{2}$$

To determine conservation priorities, the national responsibility assessment is combined with an assessment of the IUCN threat status of species [36, 48, 49] (Table 2).

An advantage of the approaches proposed in [36] is the integration of biogeographical information, such as regions defined by co-occurrence of species, distribution of large biomes, or climatic factors, into the delineation of ecological zones [7, 40] (Figs 5 and 6).

Our analysis in this example reveals that France has a high responsibility score for *Sorex coronatus* because most of the distribution of this species is within the territory of France (compare Fig 4 and Table 2). For two species (*Sorex alpinus*, *Sorex minutus*) France has a low responsibility score. *Sorex alpinus* shows preferences for alpine habitats that are less common in the focal area (France) but comprise large regions of Switzerland and Austria. For *Sorex minutus*, Russia scores very high on the responsibility metric because it includes a large segment of this widespread generalist's range. France has no responsibility for *Crocidura sicula*, which is endemic in Sicily, Italy. However, because this shrew can be found only on Sicily and a few surrounding small islands, Italy has a high national responsibility score.

## Description of the National Responsibility Assessment Tool

**Structure.** The NRA-Tool needs four predefined layers with polygons to allow assessments on a national to global scale (S1 File). The four types of basic layers are (i) files showing the geographical distribution of the focal species (FS, e.g. Fig 1), (ii) the physical geographical regions or spatial units that are either described by co-occurrence of floral and faunal elements (biogeographical unit, BU) or by a combination of bioclimatic factors (ecological unit/region; EcU) that substantially determine the distribution (Fig 5), (iii) the borders of the reference

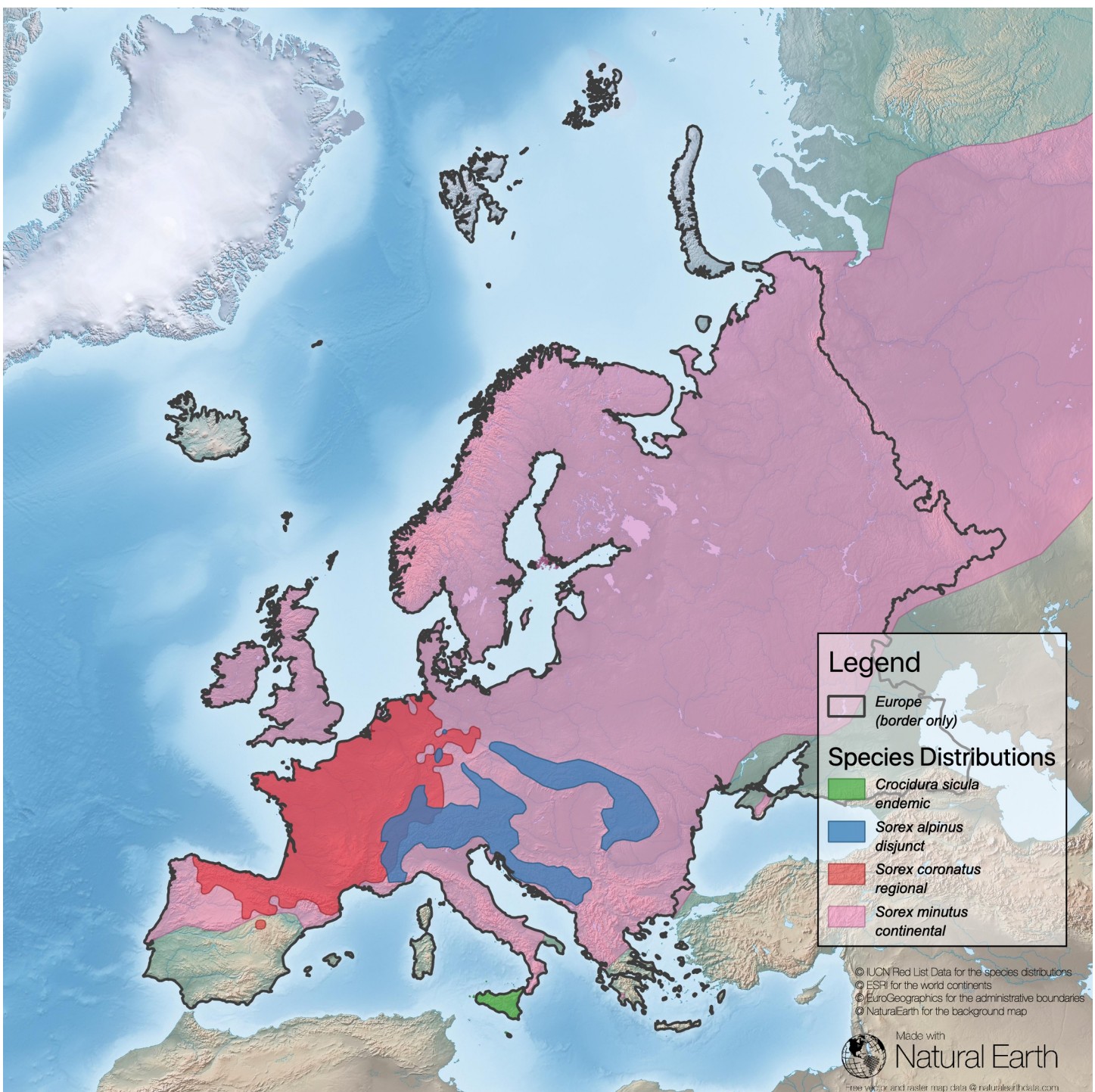

**Fig 1. Task 1 –find data on the distribution of focal species.** Here shown for four species of shrews across Europe: *Crocidura sicula* (endemic in Sicily, Italy), *Sorex coronatus* with a regional rather closed distribution, *Sorex alpinus* (disjunct distribution), and *Sorex minutus*, widely spread across Eurasia (Sources: [43–46]).

area, and (iv) the focal area(s) to be analyzed. Instead of having a separate file for the reference area, the biogeographical or ecological unit/region files can be used to define these borders. All GIS layers should be in WGS84 projection for global analyses [51] or in the same region-

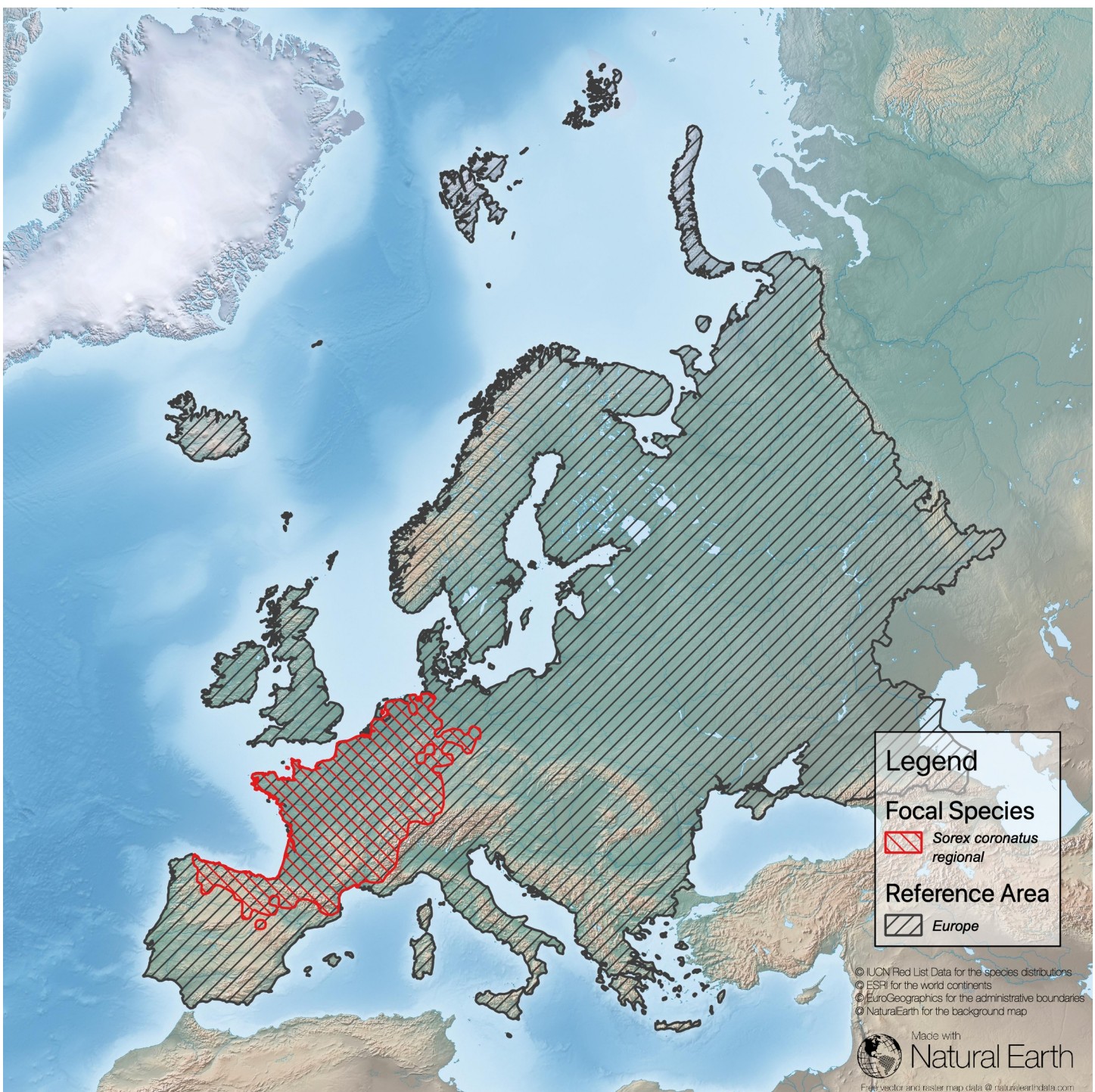

**Fig 2. Task 2 –check the suitability of the chosen reference area.** Here, the distribution area of the focal species *Sorex coronatus* is fully contained in the reference area Europe (sub-continental border is based on national borderlines; Sources: [43–46]).

specific equal area projection for analyses on a smaller scale to provide comparable results. Below, we outline the main components of the tool.

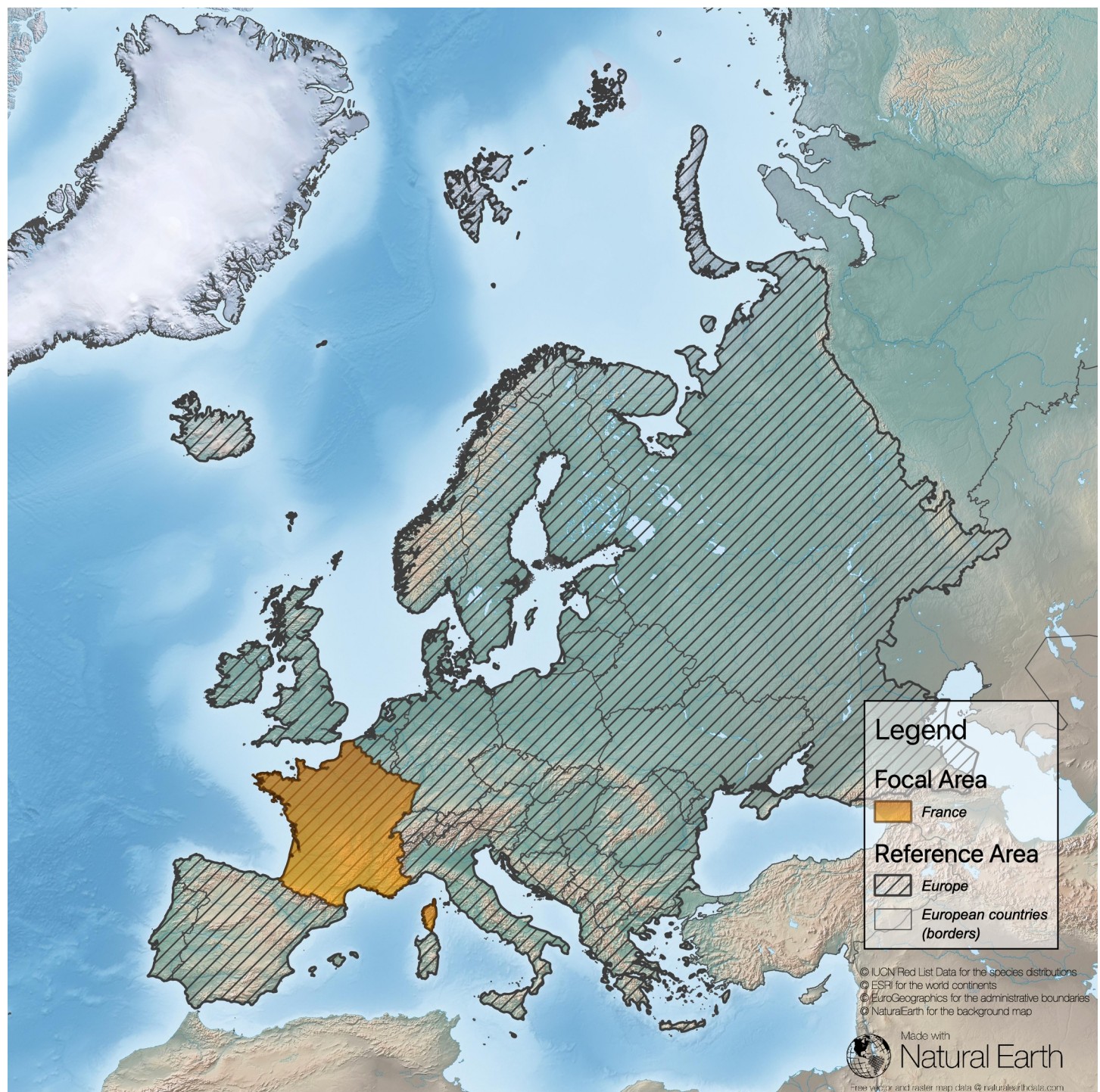

**Fig 3. Task 3 –determine the ratio of the focal area of the country and the reference area (here FA = France; European part of the territory only; RA = Europe, sub-continental border is based on national borderlines; sources: [43–46]).**

**Provision and preparation of taxonomic units or habitat units.** Analyses of regionalized responsibilities, in general, are based on an assessment of the spatial overlap between the geographical distributions of species (taxonomic unit, TU) or special habitats of conservation

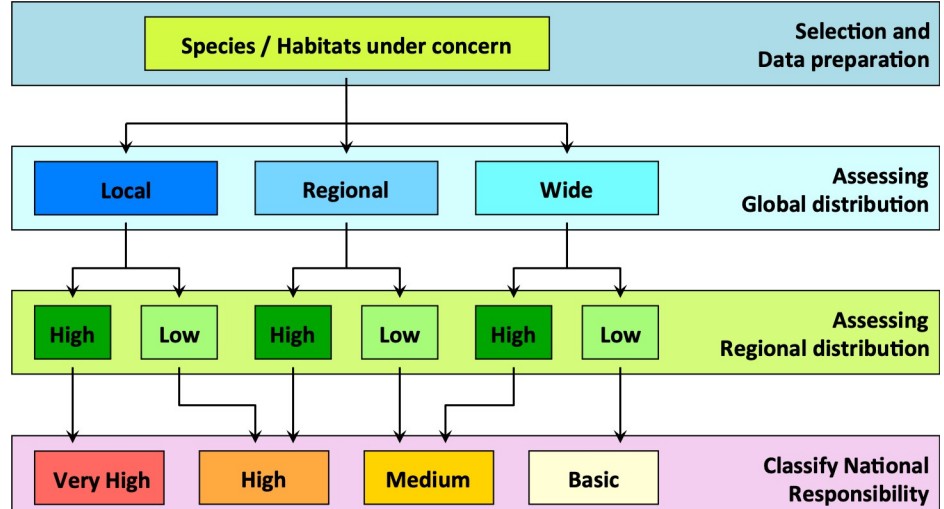

**Fig 4. Decision tree for identifying national responsibilities in species conservation after [36] (modified).** Step one is to select the taxonomic unit. Step two is to determine the distribution pattern of the species: "local" defines a species with a patchy distribution within one biogeographic region (in the case of Europe, *sensu* European Habitats Directive, Council Directive 92/43/EEC), and "wide" refers to a species distribution spanning more than one biogeographic region. The third category is "regional", wherein two-thirds of the distribution area of a species is located in one biogeographic region. Examples can be found in [37]. The final step is to calculate the proportional distribution of a species in the focal area. Two proportions are calculated–the expected distribution proportion ($DP_{exp}$) and the observed distribution proportion ($DP_{obs}$).

concern and administrative units [36, 37]. What is important is that an area can be clearly delineated as a polygon for the unit of concern (distribution area), based on available data. Examples of sources include the webservices of IUCN (https://www.iucnredlist.org/) and Map of Life (https://www.mol.org) [52]. The distribution of each species or habitat category must be available as a separate layer for use in our tool. Files downloaded from the IUCN webserver require additional processing so that data for each species is separated into a unique shapefile, and renamed by adding the prescript "s-" (following the naming convention described in more detail in the section "Technical requirements"). These shapefiles must be accompanied by an ASCII text file that contains the species names separated by semicolons (see S3 File for example).

**Focal area and reference area.** The focal area is the geographical area for which conservation priorities are determined. For national responsibility assessments, the borders of an administrative unit, such as a country or province, should be used to demarcate the focal area. The focal area must be completely enclosed by the reference area.

The reference area defines the extent and serves as a reference to determine whether the focal area contains a higher percentage of the distribution range than expected by the size of the focal area. The selection of the reference area will depend on the goal of the assessment. If the goal is to define priorities for a particular geographic region, the reference area should be the combined spatial extent of all biogeographical units in which at least one of the assessed species occurs. If the goal is to determine the national responsibility of a lower-level administrative unit, such as a province, then the reference area can be a higher-level administrative unit, such as a country. Results are comparable only within the same assessment.

All three layers described above (reference area, biogeographical unit, focal area) should contain at least one polygon. Each polygon should relate to and be sufficiently described by

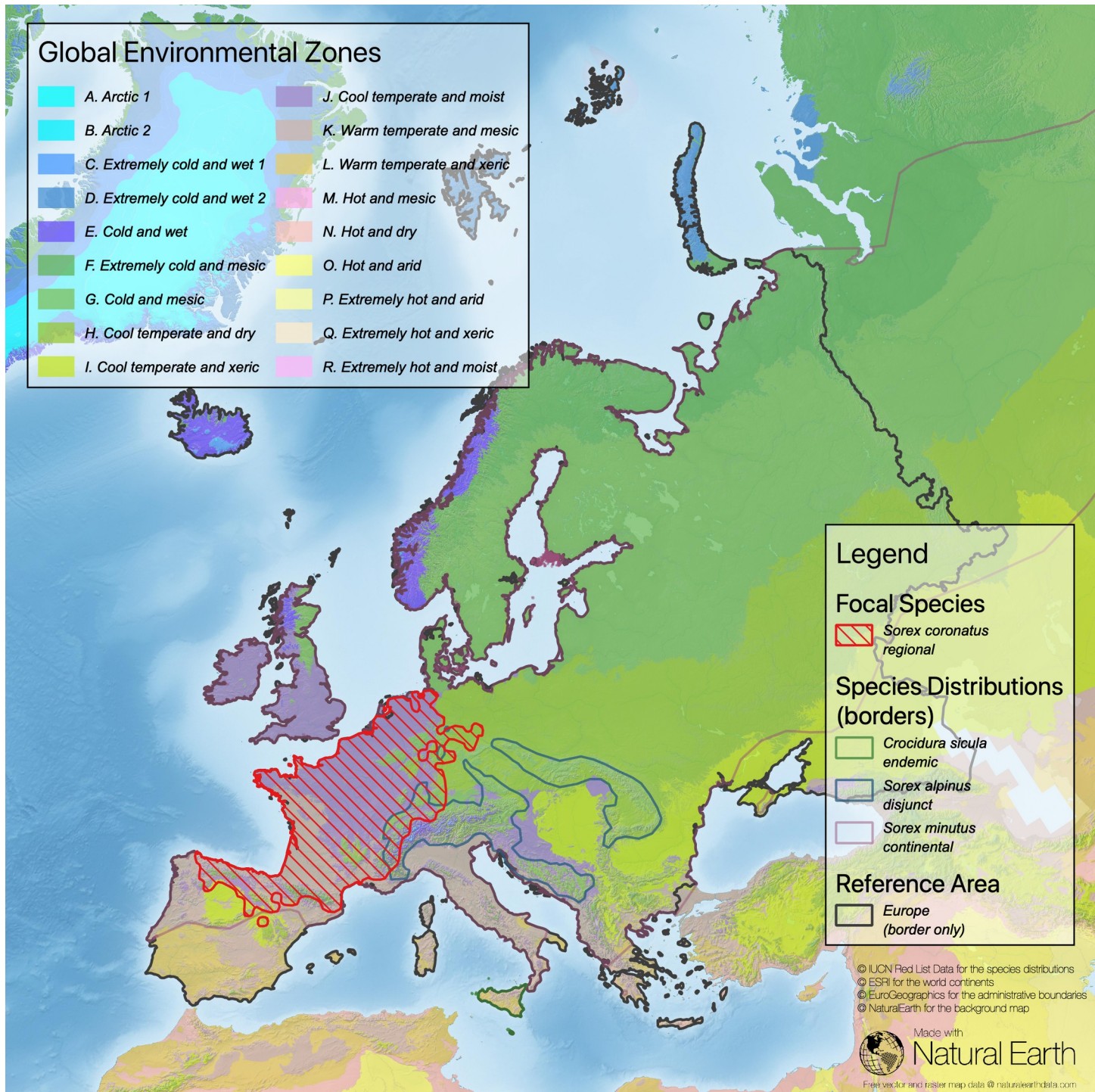

**Fig 5. Task 4 –determine the overlap between the different types of distributions of the four shrew species, the different Global Environmental Zones, and the reference area of Europe (sources: [43–46, 50]).**

one field in the table of attributes. In most cases, the biogeographical unit layer and focal area layer (e.g. countries) will have more than one unit/polygon covered by the reference area.

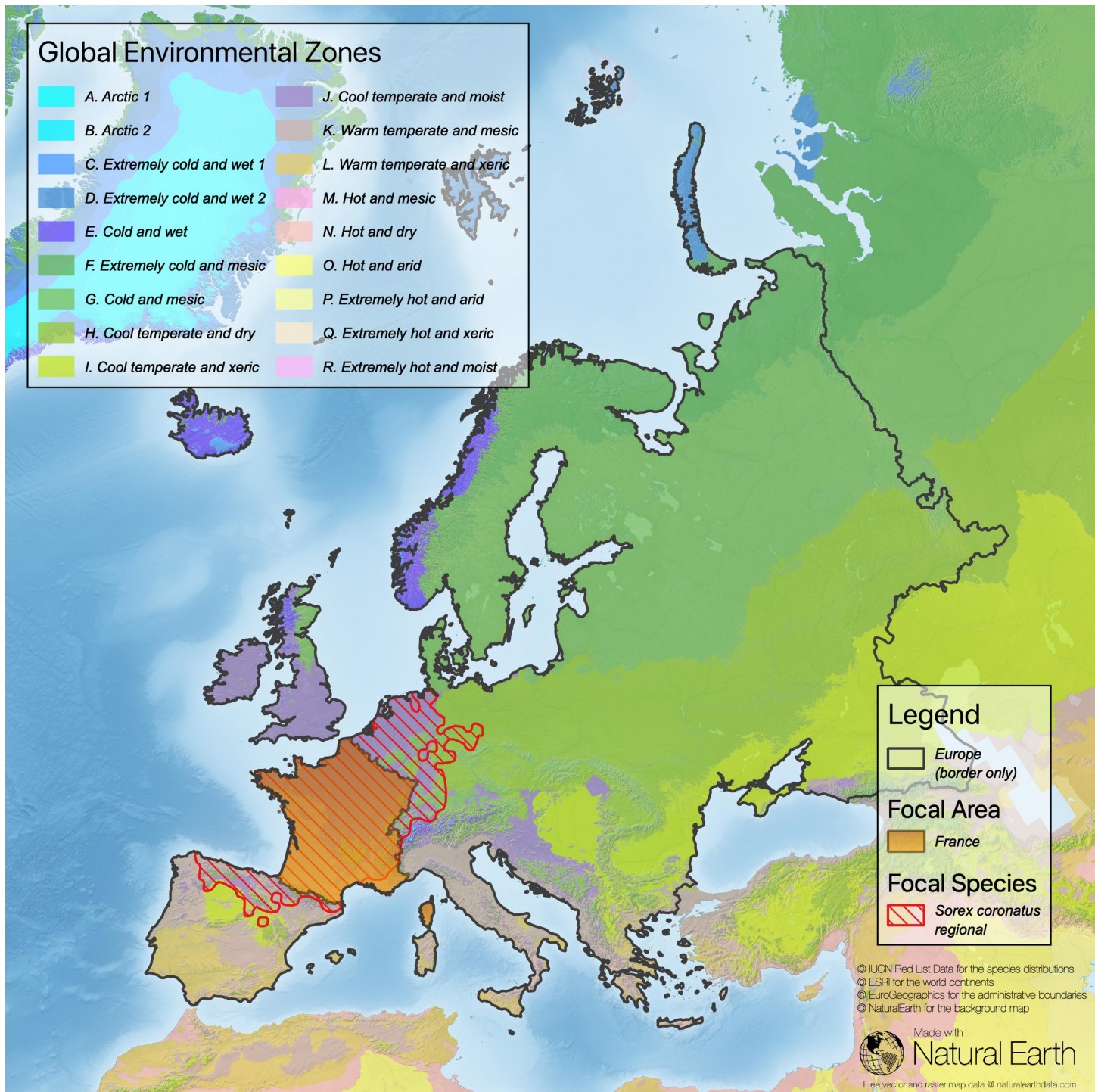

**Fig 6. Task 5 –define the ratio of the parts of the distribution of the focal species and the Global Environmental Zones contained in the focal area of the country France (European part of the territory only) and the respective parts of the distribution of the focal species and the Global Environmental Zones fully contained in the reference area of Europe (sources: [43–46, 50]).**

The biogeographical unit files we provide with our tool consist of polygons of a lower-level scale that can be grouped by a higher-level attribute. The higher-level attributes can be used to

**Table 2. Classes of conservation priorities based on scores (in brackets) for national responsibility and IUCN threat status following [36].**

| IUCN Category | National responsibility | | | |
| --- | --- | --- | --- | --- |
| | Very high | High | Medium | Basic |
| Extinct in the wild | Class 1 (25) | Class 1 (20) | Class 2 (17) | Class 2 (16) |
| Critically Endangered | Class 1 (22) | Class 2 (17) | Class 2 (14) | Class 2 (13) |
| Endangered | Class 1 (20) | Class 2 (15) | Class 3 (12) | Class 3 (11) |
| Vulnerable | Class 2 (18) | Class 2 (13) | Class 3 (10) | Class 4 (9) |
| Near Threatened | Class 2 (16) | Class 3 (11) | Class 4 (8) | Class 4 (7) |
| Least Concern | Class 3 (11) | Class 4 (6) | Class 4 (3) | Class 4 (2) |
| Data Deficient Not Evaluated | | | | |

create larger polygons by a simple dissolve procedure that aggregates all objects with the same attribute in the next highest level unit relative to the original scale. These new, enlarged polygons can then be used as natural reference areas based on biogeographical units. Alternatively, geographical units, such as continental borders, or administrative units, such as the member states of the EU, can be used for the creation of the reference area polygon.

**Assessing the distributions.** When distribution data have been obtained for the selected species, distribution areas are designated as "local," "regional," or "wide" based on the number of biogeographical units covered by the distribution area of the selected species. This step follows the recommendations of Schmeller et al. [1, 7]. Alternatively, categorization can be based on the area of overlap with biogeographical units (see S2 File for examples).

Two classifications of biogeographical units are available in the NRA-Tool for terrestrial ecosystems: The Terrestrial Ecoregions of the World (TErW, [53]) and the Global Environmental Zones (GEnZ, [50]). Both classifications subdivide their first level units (Terrestrial Ecoregions resp. Global Environmental Zones) into sub-regions of a lower geographical extent based on additional regional criteria. Following arguments provided by Schmeller et al. [1], we strongly suggest using GEnZ [50, 54] as biogeographical units for implementation at the continental and global scale. Notably, the NRA-Tool also provides a spatial assessment of conservation priority in the form of maps and associated attribute tables for all focal areas.

Our tool is not limited to analyses of terrestrial ecosystems. The architecture allows for any ecosystem classification, provided the layers consist of a complete set of spatially inclusive and comprehensive polygons fully covering the reference area. Thus, the NRA-Tool can be used to perform global assessments of species living only in freshwater ecosystems or even species that are exclusively marine [55, 56].

**Workflow.** The workflow (Fig 7) starts with the selection of the taxonomic unit (TU) to be analyzed from the dataset of all taxonomic units. The distribution pattern is then classified as "local," "regional," or "wide" based on the shapefiles of the reference area and the focal areas, and the distribution of the taxonomic unit with respect to the focal species. In the second step, the NRA-Tool calculates the expected and observed distribution areas and determines national responsibility scores and conservation priorities for the focal species as described above. As an output, conservation priority classes are assigned to all or selected countries or regions for the focal species. The results are provided as maps (shapefiles) with respective attribute tables containing the values of national responsibility scores and/or conservation priority of the assessed focal species. The tool also provides the same information and a short summary in an ASCII text file that can be stored digitally for further analyses or visualization.

The NRA-Tool offers two options for classifying distribution patterns: the Polygon Count-Approach (PC-A) [1, 7] and the Polygon Area-Approach (PA-A). The difference between the Polygon Area-Approach and the Polygon Count-Approach option is the way biogeographic

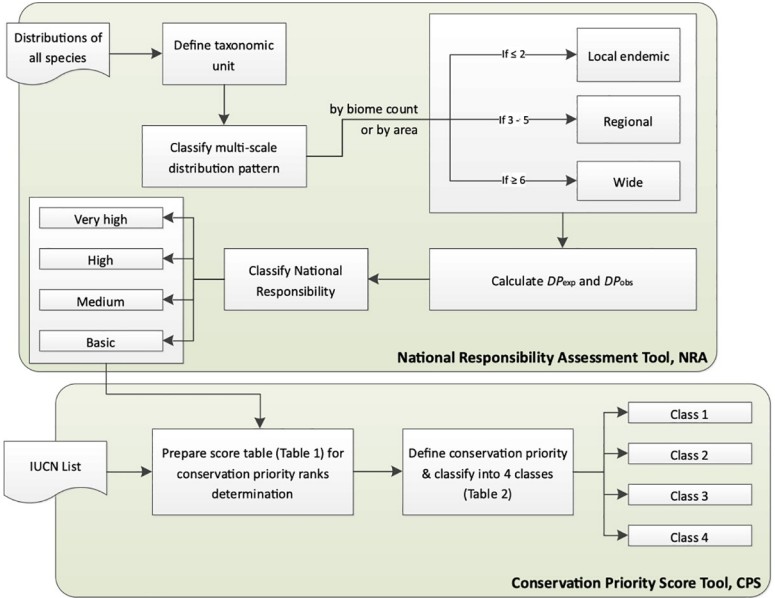

**Fig 7. The workflow of the National Responsibility Assessment Tool (IUCN: International Union for Conservation of Nature; NR: National Responsibility; CP: Conservation Priority; $DP$exp: expected distribution proportion, and $DP$obs: observed distribution proportion).**

units are related to the focal species distributions and focal areas in regard to the determination of distributions of selected species as "local", "regional", or "wide" (for examples see S2 File).

In the Polygon Count-Approach, the number of biogeographical units covered by the distribution of the focal species is used to determine whether its distribution is "local," "regional," or "wide," (Fig 7). These class borders are flexible and can be changed to other values for further analyses. However, we strongly recommend using the standard setting that is described in [1, 7].

Using the count of biogeographical units can be misleading, especially in cases where the environmental conditions change rapidly over space and the overall area of biogeographical units overlapping with the focal area of an endemic species is small. In such cases, it is common to find three or more biogeographical units within a very small area, when only one biogeographical unit might be found in a similarly sized area in a region with less spatial variability, for example. The alternative approach implemented in the NRA-Tool calculates the real area proportion of the various biogeographical units that overlap with the distribution area of the focal species (S1 and S2 Files). To avoid biased area calculations, this method requires data with adequate spatial resolution in the correct geographical projection. This approach also requires detailed analyses of the resulting statistical distributions of the proportional area by which each biogeographical unit overlaps the distribution of the focal species. The Polygon Area-Approach is still in experimental stage, and most results published so far have used the Polygon Count-Approach with biomes or environmental zones [50]. We recommend using the Polygon Count-Approach (PC-A, 'by Biome count' option in the menu) described in [1, 7] in calculations for practical applications.

With this manuscript, we provide four appendices containing supporting information on the program and its functioning. S1 File illustrates the "User interface", S2 File gives an example for the Polygon Count-Approach and Polygon Area-Approach calculations, S3 File

provides the list of species used in the analyses and S4 File describes the performance and explains opportunities for optimization.

**Technical requirements.** The NRA-Tool is a module that can be added to the ArcGIS Toolbox. It was developed in ArcGIS 10.x and C# [57] using Visual Studio 17 [58] with.NET 4.6.1 [59], ArcGIS runtime SDK for.NET [60], and ArcObjects SDK [61].

Files to be used with the NRA-Tool must be named according to the following convention: The species distribution file names must start with a lowercase "s" followed by a hyphen. After the hyphen, the name can be chosen freely (e.g., "s-vulpes_vulpes.shp" for the red fox). For (global) ecological units or reference areas, file names must start with a lowercase "g" followed by a hyphen (e.g., "g-biomes.shp" and "g-border_of_asia"). Focal area file names must start with a lowercase "f" followed by a hyphen (e.g., "f-asian_countries.shp"). Files not named according to these conventions will not be displayed and therefore will not be selectable in the tool.

In ArcGIS, geodata should have a spatial reference if they are used for spatial analyses based on calculations of distance and area [51]. To avoid problems and erroneous results caused by incompatible projections, all input files for the NRA-Tool should use the same ArcGIS geore-ferencing and projection system. Generally, we recommend WGS84 for global analyses. Analyses on continental or smaller scales should be performed with more appropriate conformal area projections where polygon areas represent country areas. This is especially critical when using the Polygon Area-Approach. Other sources of bias must be considered when using the NRA-Tool, including the use of different formats, and the spatial resolution of data [1, 36]. Accordingly, we recommend that users check the georeferencing, projections, and resolutions of the input shapefiles as well as the CPU, RAM, and disk space of their computer system prior to using the NRA-Tool.

The computation times for national responsibility and conservation priority assessments are highly dependent on computer performance and are affected by the limitations of the operating system and hardware, as well as the spatial data input. For example, when calculations were made for 58 countries and regions on an area-only basis (i.e., comparison of areas without considering the shape, spatial extent, and overlap of the different units of concern) using a laptop with a 1.8 GHz processor and 4 GB RAM, the total computation time for the analyses of national responsibility and conservation priority for one species (*Cyornis rubeculoides*) based on TErW [53] was approximately 8 seconds, and based on GEnZ [62] was approximately 50 seconds Despite the longer runtime, when using complex, high-resolution spatial and environmental data to compute distribution range data for many species, we recommend using the GEnZ shapefile downloadable from [62] and applying the methods in [36, 37]. This task is computationally demanding and the runtime may be many hours or even days, especially if the analyses consider the whole globe as the reference area.

**Further development.** In addition to the development and improvement of the ArcGIS tool, we have started the development of a similar tool as an extension for the Open Source GIS QGIS [63], which will be made available as Open Source in future. The Polygon Area-Approach currently only uses information about the range/extent of the species distributions and should be expanded to allow for the use of abundance data. Additionally, it might be possible to integrate more options for particularly spatially-oriented conservation priority assessments with similar approaches, e.g. [31].

**Availability of software and data.** The NRA-Tool can be downloaded directly from [64, 65] or the interactive representation of the SCALES-project [66] called SCALETOOL [66–68]. The GEnZ can be found and downloaded from [62]. The publicly available species distribution dataset for selected shrew species is derived from IUCN Red List data and can be downloaded from [44] (not for commercial use). World continent data are available from ESRI [43], while

the geographical data for administrative units (country areas, borderlines) from EuroGeographics can be downloaded from [45]. For the background map we used cross-blended hypsometric tints with relief, water, drains, and ocean bottom in scale of 1:10 million that is publicly available under the Creative Commons license from Natural Earth [46].

**Case studies of bird species in Asia by Global Environmental Zones counts and Global Environmental Zones area.** Here, we selected 58 Asian countries and regions as the focal areas or reference areas and categorized them according to their biogeographic characteristics: (1) tropical or temperate climate, (2) small or large area, (3) island or mainland. Birds that are (1) widespread and (2) migratory are considered too, although their distribution ranges exceed or lie outside of the reference area, respectively. However, it is not the aim of this study to discuss special aspects of national responsibility assessments related to migration or animal movement; our purpose is simply to demonstrate our tool, using this dataset as an example. We determined the national responsibility scores and conservation priorities for 258 bird species (from BirdLife International http://www.birdlife.org) in the 58 Asian focal countries (see S3 File). Nine of the bird species were vulnerable (VU); eight were near-threatened (NT), and 241 were of least concern (LC).

We performed two analyses, both of which used GEnZ [50], but differed by the choice of either the 'By biome-area' option in the panel (Polygon Area-Approach) or the 'By biome count' (Polygon Count-Approach) option. Both approaches use the polygons of the chosen map as the ecological units/regions (GEnZ in our case).

S1 Data gives an example for the content of the output files (dBase) for the results based on the Polygon Count-Approach and the Polygon Area-Approach shown in Figs 8 and 9. S2 Data provides a comparison of results for countries having medium national responsibility for at least one out of 258 species studied. The authors are solely responsible for the content and functionality of these materials. Queries (other than absence of the material) should be directed to the corresponding author.

## Results

Using the Polygon Count-Approach two bird species were classified as "local" (locally distributed), six as "regional" (regionally distributed), and 250 as "wide" (widely distributed) according to Schmeller et al. [1, 36] (Table 3, S1 and S2 Data).

### National responsibility

Our assessment of national responsibility shows that only a few countries, most in the south of Asia (China, Laos, Brunei, Cambodia, India, Indonesia, Malaysia, Myanmar/Burma, Thailand, Vietnam, and Taiwan), have high to very high national responsibility scores for the analyzed bird species (Fig 8A and 8B) with China responsible for the highest number of species. The ranking of the countries in the list corresponds to the number of species for which they have high and very high national responsibility scores.

Thirteen countries (Myanmar, Vietnam, Nepal, Bhutan, Laos, China, Thailand, India, Cambodia, Bangladesh, Malaysia, Indonesia, Taiwan; Fig 8C and 8D) have medium national responsibility scores for at least 40 species, which represents the 75%-quartile of the distribution of the number of species for which a country is responsible (see Table 3, S2 Data). The countries are ranked according to the number of species for which they have medium national responsibility. Comparing the results produced by the two approaches available in our tool, we found some differences. When we used the PA-A instead of the PC-A, no country gained more than four additional species and the maximum number of species for which a country lost medium responsibility was 10. However, over all countries the net change was zero

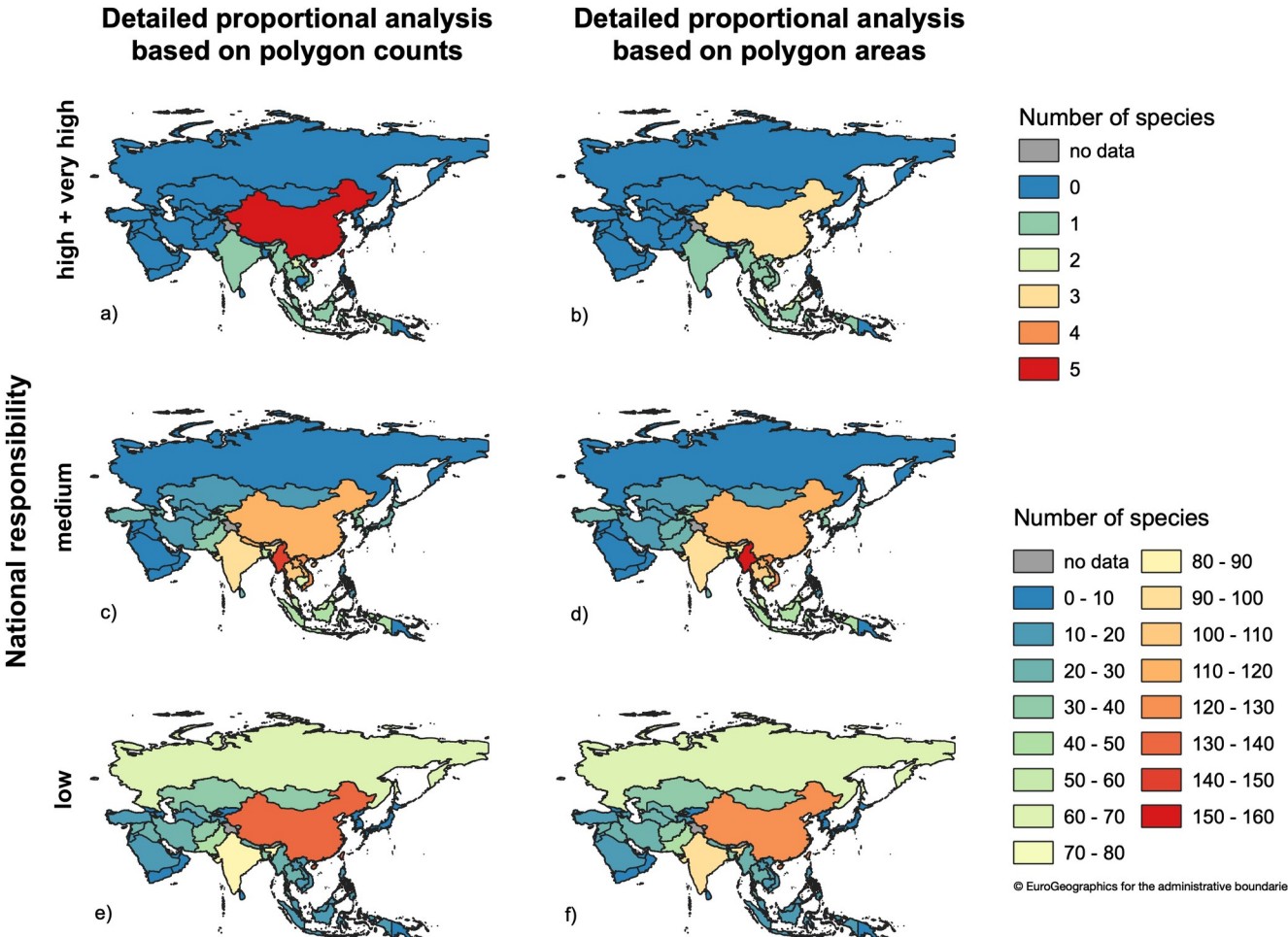

**Fig 8. Number out of 258 analyzed species in Asian regions and countries for which a country has a certain national responsibility.** BUs = GEnZ.

(arithmetic mean = 0.13, median = 0). Looking at rank switching above the 75%-quartile, we saw only one exchange, which occurred between Nepal and Bhutan (2 out of 13 = 15.38%). When we used the PA-A instead of the PC-A, Bhutan had a medium responsibility score for three additional species. This is in stark contrast to the rank switches we observed below the 25%-quartile of the distribution. Here, we found that 10 out of 13 (76.92%) countries changed rank, which shows the summation of rank changes over larger ranges and in different positions, but not paired exchanges in the same positions (Table 3).

Generally, large countries like Russia, China, and India receive low national responsibility scores for a high number of species (Fig 8E and 8F).

## Conservation priority

Using the Polygon Area-Approach with GEnZ [50] as biogeographical units and IUCN global Red List status to determine conservation priority, none of the 258 bird species in the focal area fell within Class 1 (Fig 9A and 9B), due to their low threat status. Generally, the results for both approaches were similar for national responsibility as well as for conservation priority (Fig 9). However, using the Polygon Area-Approach based on intersections of the overlapping polygons leads to results more balanced among countries compared to the simple Polygon Count-Approach originally used in [1, 7, 36, 37].

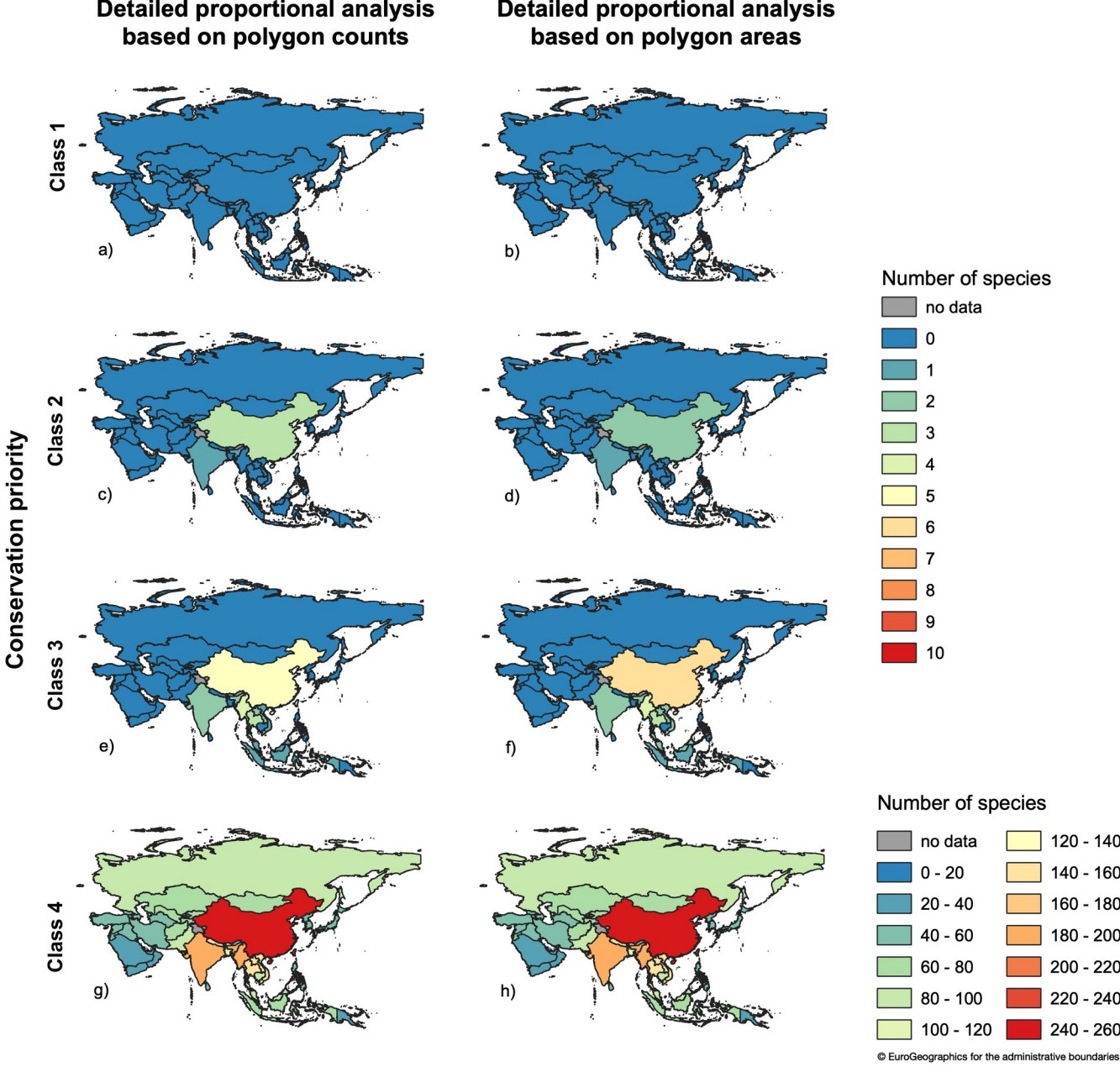

**Fig 9. Conservation priorities for the number out of 258 analyzed species in Asian regions and countries that fall in a certain conservation priority class.** BUs = GEnZ.

## Discussion

The standardized determination of national responsibilities and conservation priorities across regions with multiple jurisdictions and administrative borders is an important step to advance and coordinate international conservation efforts. Here, we presented a software tool to assist in the determination of national responsibilities and conservation priorities. We showcased our tool with an example dataset of 258 Asian bird species. Our tool improves geoprocessing procedures for calculating national responsibilities and conservation priorities using a large

**Table 3. Summary statistics for the number of species per country and the ranking of countries based on the two approaches, the Polygon Count-Approach and the Polygon Area-Approach, and stored in the results tables created by the ArcGIS NRA-Tool (see S1 and S2 Data).**

| | | Number of Species | | | | | | | |
|---|---|---|---|---|---|---|---|---|---|
| | | Mini-mum | 25% Quartile | 50% Quartile (Median) | 75% Quartile | Maxi-mum | Arithmetic Mean | SD |
| Species per Country | PC-A | 1 | 10.75 | 21.00 | 38.25 | 150 | 36.17 | 38.12 |
| | PA-A | 1 | 10.50 | 20.50 | 39.75 | 152 | 36.31 | 38.73 |
| | Thresholds (rounded) | 1 | 11 | 21 | 40 | 152 | | |
| | PA-A—PC-A | -6.00 | -0.25 | 0.00 | 1.00 | 4.00 | 0.13 | 1.65 |
| | **Range** | | **0–25%** | **25–50%** | **50–75%** | **75–100%** | **Overall** | |
| Country Rank | PC-A = PA-A (true) | | 3 | 4 | 5 | 11 | 23 | |
| | PC-A <> PA-A (false) | | 10 | 10 | 7 | 2 | 29 | |
| | Sum (true+false) | | 13 | 14 | 12 | 13 | 52 | |
| | Percentage (true) | | 23.08% | 28.57% | 41.67% | 84.62% | 44.23% | |
| | Percentage (false) | | 76.92% | 71.43% | 58.33% | 15.38% | 55.77% | |

spatial dataset. The tool also provides users with an interface and attractive visual outputs to facilitate the inclusion of national responsibility and conservation priority assessments in national and regional biodiversity reports. It is also possible to use it in combination with software tools for conservation prioritization assessments based on different algorithmic foundations like Marxan *with Zones* [17] and Zonation [16, 15], or multivariate statistical analyses [33]. These avenues of integration ensure that our tool can provide important assistance in making informed policy decisions [1].

Our results show that the national responsibility assessment is sensitive to the size of focal areas and focal species ranges when area and biophysical regions, such as the GenZ, are used to classify focal species distributions. Both global and regional assessment scales [36] are strongly dependent on spatial distributions, the extent of overlap between focal species distribution and focal area, and the number of biogeographical units in the focal area. A more limited global species distribution is likely to result in a smaller proportion of overlap between biogeographical units and the reference area. At the same time, the proportion of overlap between the focal species distribution and the focal area will probably be high. Accordingly, regional responsibility is a biogeographic metric that is related to the distribution range [69]. If a species is widespread outside of a region of interest, then, for a given reference area, the regional responsibility score for the species in the focal area will be low [69].

Maps of national responsibility and conservation priority obtained from our data using the Polygon Count-Approach and the Polygon Area-Approach showed that countries with at least "medium" responsibility for a high number of species fall into conservation priority Class 2. The Russian Federation did not have a high responsibility score for any species in the regional assessment (see Fig 4). This is because Russia shares many widespread species with Europe, has few endemic species relative to its size, and covers fewer Global Environmental Zones compared to other large counties, such as China, for example.

We found that countries previously identified as Asian biodiversity hotspots [70, 71] had higher responsibility scores for a larger number of species compared with countries outside designated hotspot regions. According to Pimm et al. [70], these hotspots overlapped by 68% with BirdLife International's Endemic Bird Areas, 82% with the areas designated as International Centres of Plant Diversity and Endemism by IUCN/WWF, and 92% with the most

critically endangered eco-regions on the WWF/US Global 200 List. The assessment of national responsibility and conservation priorities allowed us to perform further analysis and mapping using attribute tables. For example, in our case study we obtained the number of species associated with a "very high" and "high" level of national responsibility for multiple focal species in each focal area. The national responsibility approach, therefore, provides additional policy-relevant information by identifying those species for which a country has high to very high responsibility scores within biodiversity hotspots. With this information, decision makers can allocate limited resources to the most urgently needed protection measures, including capacity-building [1, 72].

For more in-depth analyses and interpretation, the tool provides the results not only in map form but also in tables containing all input information, calculated areas, and resulting scores (Table 3, S1 and S2 Data). These outputs can provide insight into the discrepancies between the results of the two methodical approaches offered by our tool. These two approaches should be compared and evaluated in further methodological studies focusing on statistical effects and the functional relationship between environmental variability across countries and value distributions.

Our own analysis shows that the more advanced PA-A provides more differentiated results than the PC-A because it takes area of overlap into account rather than simply using the number of overlapping polygons. This leads to differences in weighting, especially when large countries have relatively little overlap with focal and reference areas. The size of a country also plays a role when considering the number of species for which basic responsibility must be assumed. The larger the territory of a country, the greater the probability that it will include significant portions of the range of a species in the corresponding reference area—a species for which the country then bears basic responsibility.

The NRA-Tool described here will allow the determination of national responsibility and conservation priorities across large geographic scales and across all species (and habitats) for which distribution data are available. With the NRA-Tool we are able to conduct global analyses and to inform global processes, such as IPBES assessments, about national responsibilities of each member state and to make suggestions on conservation priorities for different species. It can also be used to identify information gaps resulting from a lack of monitoring programs that target species for which countries have a "high" to "very high" degree of responsibility [72]. The results provided by the NRA-Tool, which are hierarchical lists, can be used to start capacity-building efforts in less surveyed regions and countries. The results can also be used in the various other ways outlined earlier [1]. With this information, policy and decision makers will be better equipped to assess when and how policy should be adapted, reinforced, or developed to fulfill the Convention on Biological Diversity's Aichi targets. The maps and tables produced with the NRA-Tool are visual supports for making decisions about international biodiversity conservation.

## Supporting information

**S1 File. NRA-tool user interface.**
(PDF)

**S2 File. An example for the calculations of the Polygon Count-Approach and the Polygon Area-Approach.**
(PDF)

**S3 File. List of species used in the illustration example.**
(ZIP)

**S4 File. Performance and possibilities for optimization.**
(PDF)

**S5 File.**
(ZIP)

**S1 Data. Example for the content of the output files (dBase) for the results based on the Polygon Count-Approach and the Polygon Area-Approach shown in Figs 8 and 9.**
(ODS)

**S2 Data. Comparison of results for countries having medium national responsibility for at least one out of 258 species studied.**
(ODS)

## Acknowledgments

We would like to thank BirdLife International (http://www.birdlife.org) for provision of geographical distributions of 258 bird species, The IUCN Red List of Threatened Species™ [44] for the provision of geographical distribution for 4 shrew species, ESRI [43] for the publicly available continental borders, EuroGeographics [45] for publicly available data of administrative borders, and Natural Earth [46] for the background map. We thank the two anonymous reviewers, whose comments / suggestions have contributed significantly to the improvement and clarification of this manuscript, and Alessandra Moyer for linguistic proofreading.

## Author Contributions

**Conceptualization:** Yu-Pin Lin, Dirk S. Schmeller.

**Data curation:** Yu-Pin Lin, Tzung-Su Ding, Yung Chieh Wang, Wan-Yu Lien, Reinhard A. Klenke.

**Formal analysis:** Yu-Pin Lin, Tzung-Su Ding, Yung Chieh Wang, Wan-Yu Lien.

**Funding acquisition:** Yu-Pin Lin, Klaus Henle.

**Methodology:** Yu-Pin Lin, Dirk S. Schmeller, Tzung-Su Ding, Klaus Henle, Reinhard A. Klenke.

**Project administration:** Yu-Pin Lin, Klaus Henle, Reinhard A. Klenke.

**Resources:** Yu-Pin Lin, Reinhard A. Klenke.

**Software:** Yu-Pin Lin, Reinhard A. Klenke.

**Supervision:** Yu-Pin Lin, Dirk S. Schmeller, Tzung-Su Ding.

**Validation:** Dirk S. Schmeller, Yung Chieh Wang, Reinhard A. Klenke.

**Visualization:** Yu-Pin Lin, Wan-Yu Lien, Reinhard A. Klenke.

**Writing – original draft:** Yu-Pin Lin, Dirk S. Schmeller, Reinhard A. Klenke.

**Writing – review & editing:** Klaus Henle, Reinhard A. Klenke.

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
