## [Decision Letter · Decision Letter 0]

21 Apr 2020

PONE-D-20-05865

A GIS-based policy support tool to determine national responsibilities and priorities for biodiversity conservation

PLOS ONE

Dear Dr. Klenke,

Thank you for submitting your manuscript to PLOS ONE. After careful consideration, we feel that it has merit but does not fully meet PLOS ONE’s publication criteria as it currently stands. Therefore, we invite you to submit a revised version of the manuscript that addresses the points raised during the review process.

Two experts in the field have reviewed the paper, who have provided detailed and constructive comments. Although both they find the study interesting, they raise a number of questions and concerns with it, so I am recommending that you undertake a major revision of your manuscript.

Overall, both reviewers see the need of providing information about the conceptual framework on which is based this GIS tool. I agree that, in general, the manuscript is hard to follow because of numerous acronyms and many taken-for-granted concepts. Although the concept of national responsibility in spatial prioritization approaches has been defined and explained previously in different papers, authors should make an additional effort to explain the basics of this framework in order to make clear how this tool works and adapt the description of the software to those readers that are not familiar with this framework. Otherwise, the reader can easily get lost.

I invite you to carefully respond to the reviewers' comments and revise your manuscript accordingly. Your manuscript will be sent for a second round of revision, and it is therefore imperative you provide thorough responses/revisions to each of the comments and suggestions below.

Additionally, please make sure that the manuscript meets PLOS ONE criteria for manuscripts that describe new software for applications. Specifically these reports must meet the criteria of utility validation and availability which are described in detail at http://journals.plos.org/plosone/s/submission-guidelines#loc-methods-software-databases-and-tools. Please, in your response letter explain how your manuscript meets these criteria.

We would appreciate receiving your revised manuscript by Jun 05 2020 11:59PM. To enhance the reproducibility of your results, we recommend that if applicable you deposit your laboratory protocols in protocols.io, where a protocol can be assigned its own identifier (DOI) such that it can be cited independently in the future. For instructions see: http://journals.plos.org/plosone/s/submission-guidelines#loc-laboratory-protocols

We look forward to receiving your revised manuscript.

Kind regards,

Pedro Abellán

Academic Editor

PLOS ONE

Journal Requirements:

Reviewers' comments:

Reviewer's Responses to Questions

**Comments to the Author**

1. Is the manuscript technically sound, and do the data support the conclusions?

Reviewer #1: Yes

Reviewer #2: Yes

2. Has the statistical analysis been performed appropriately and rigorously? 

Reviewer #1: Yes

Reviewer #2: N/A

3. Have the authors made all data underlying the findings in their manuscript fully available?

Reviewer #1: Yes

Reviewer #2: Yes

4. Is the manuscript presented in an intelligible fashion and written in standard English?

Reviewer #1: Yes

Reviewer #2: Yes

5. Review Comments to the Author

Reviewer #1: This manuscript regards the application of a prioritization method together with the national responsibility approach (an approach already used in several scientific works) in a novel GIS tool. This tool is presented with data from Asia. The manuscript is well written but there are a number of major concerns that need to be addressed.

Introduction

I think that national responsibility and conservation priority approaches should be thoroughly presented. What is national responsibility? How can this be derived? This in addition to the specific explanation given in the method section. There are a number of prioritization methods (including or not responsibility) applied in the literature and a brief overview should be given (see also indications in the discussion section). Data availability and type can influence the applicability of spatial prioritization approaches. This should be introduced as it can be relevant for application of the tool to other regions.

Methods

There are some aspects that deserve further explanation. Please give more details on what is meant with focal and reference area (I don’t think referring to Schmeller and colleagues suffices). Regarding the additional factor in the prioritization process, I miss how the different scores and classes were decided and assigned.

I think that a graphical framework of the National Responsibility Assessment Tool would enable a better understanding of the type of data, steps and requirements needed for its application (maybe integrated in fig. 1).

L119-120: I suggest anticipating that this can be used at the global as well as at other scales (regional? national?).

L126-127: I do not understand what files authors refer to. The message of this sentence is not clear.

L127: “All layers resp. GIS” is not clear.

I found really complex to follow the “Focal area and reference area” section. A table could help in reporting what are the different options. One additional suggestion is to add the factors used in the applied example (see comment on S1). This would enable also to use as example specific species. It is not clear whether species data can be points, polygons and/or based on grids.

Even though I do not have a technical background regarding ArcGIS, there is one important point from my side. This tool works in ArcGIS that is a private platform. There are many other programmes such as QGIS and R statistics that are open source and can be accessed without any restrictions. Why didn’t authors decided to produce tools or scripts for open source programmes? I think this would help the application of the approach.

L259-260: It is not clear if one single biogeographic category is assigned to each country.

L260-261: Which scientific reference is used to identify widespread and migratory species?

Fig. 3: I do not think this map is relevant to understand the proposed process or to show its results! This could be part of one of the Supplementary files.

Overall I think that acronyms do not help the reader.

Results

L273: Are the ranges of the number of biogeographic zones used to differentiate between local, regional or widely distributed species fixed? If these are fixed, as I have understood, such ranges should be reported in the method section and removed from the results.

Fig. 4: Not clear what the numbers reported in the figure legend mean. The caption misses this information and what the different colours mean.

Fig. 5: Not clear what the numbers reported in the figure legend mean. The caption misses this information and what the different colours mean.

Discussion

L308-309: Friendly and attractive environment! I think authors can state this once they have users' feedback.

Results for the study case show that in some cases there can be a strong dependency on the national responsibility category (i.e. on species distribution) as many species usually are classified as least concerned. A discussion of this and other possible limitations should be integrated in the discussion.

L323-328: It would be nice to have some results on this overlap in the supplementary material.

I suggest to highlight the possibility or not of adding some indicators of conservation priority. For example, current and future acting forces can help to set priorities also in terms of conservation and management activities. There are a number of works that use these factors to identify priorities (e.g. examples from different continents: Zhang et al. 2014; Campagnaro et al. 2018; Carvalho et al. 2020). These works can be used to make additional comments on the need of spatial information and compare different proposed tools. These works can also help in highlighting possible integrations to the spatial tool that authors are presenting.

Campagnaro, T. et al. (2018). Identifying habitat type conservation priorities under the Habitats Directive: Application to two Italian biogeographical regions. Sustainability, 10(4), 1189.

Carvalho, F. et al. (2020). Methods for prioritizing protected areas using individual and aggregate rankings. Environmental Conservation, 1-10.

Zhang, L. et al. (2014). Determination of priority nature conservation areas and human disturbances in the Yangtze River Basin, China. Journal for nature conservation, 22(4), 326-336.

Supporting material

S1: I think that some examples can be reported in the main text (see comments on the method section). For example, “Reference Area: Asia border” and “Focal Area: Asian countries”. This would benefit comprehension of the national responsibility tool.

S1 L18-20: I suggest reporting these possible issues in the main text.

S2 L12: Small typo: “CNTRY_NAME”.

S3: Are the reported examples of possible cases already presented in other publications? I find this supplementary extremely useful to better understand possible cases.

Reviewer #2: Thank you for the opportunity to review this manuscript. This study develops and demonstrates a GIS-based tool, which can be used to produce National Responsibilities and Conservation Priorities for multiple species using freely available data.

For me, this study demonstrates a great tool, which is relatively easy to use and has a pretty good flexibility to be applied on different scales and locations. Furthermore, using freely available species distribution and ecoregion data and setting a standardized assessment for countries are the strengths of this tool. In my opinion, this can certainly contribute to setting conservation prioritizations for policy makers.

Although I know the focus of this study is about the tool itself, I have a question about the method behind this tool, which is the issue about wide-ranging nomadic species and migratory species.

Could this method cause an issue like “tragedy of the commons” for wide and evenly distributed species? For example, would wide-ranging nomadic species be considered as basic in NR throughout all countries it lives? How would you address this issue in the method?

In Runge et al. (2015), they suggest that it might be more suitable to use the minimum range nomads occupy across multiple years because of their fluctuated distribution. The minimum range might contain some climate refugial sites or high quality habitats for those wide-ranging species.

Runge et al. (2015) Geographic range size and extinction risk assessment in nomadic species. Conservation Biology

For migratory species, I’m curious about how you deal with them in the study. Would you mind to provide some information about this in the manuscript? since this study also includes several migratory species. Do you use their breeding ranges, non-breeding ranges, or passage ranges, or all combined, or all but separated (which means one migratory bird might be counted as NR and CP in countries where it breeds, winters, and passes by). I suppose the last method, which is using all ranges separately, make more sense because migratory populations rely on all habitats for different life stages. However, some migratory birds in Asia might have their non-breeding ranges or passage ranges extend to Oceania (e.g., Australia and New Zealand)

This may influence your results (Fig. 4, and 5) because the proportion of migratory birds among all avian communities can be very high in the north hemisphere. See Somveille et al. (2013) Mapping Global Diversity Patterns for Migratory Birds. PLOS ONE.

Minor comments as follows

line 32: to prioritize conservation what? to prioritize conservation resources? funds? effort? actions?

line 54: could you provide one or a few sentences in the introduction to define or explain “national responsibility”? If I didn’t read any previous studies about NR, I would only realize what it really is until the Methods…

line 184: While these two BUs may be useful for terrestrial species, incorporating “Freshwater Ecoregions of the World” might further enlarge the applicability of this tool. The distribution of freshwater biodiversity may be more relevant to the shape and size and boundary of watersheds/river basins than local climate or terrestrial vegetation. The freshwater ecoregion data is also freely available online. See Abell et al. (2008) Freshwater ecoregions of the world: A new map of biogeographic units for freshwater biodiversity conservation. BioScience.

line 312: Which part of your results supports this statement about “sensitive”? Is there any quantified sensitivity? such as sensitivity analysis or a plot about the correlation between the range size of FS and biophysical regions and NR? or simply based on the description.

line 528: “probability” or “responsibility”?

6. PLOS authors have the option to publish the peer review history of their article (what does this mean?). If published, this will include your full peer review and any attached files.

Reviewer #1: No

Reviewer #2: No

---

## [Author Response · Author response to Decision Letter 0]

21 Sep 2020

We have overhauled most of the sections in our text based on your and the reviewers comments, provided more detailed information about the conceptual framework on which the ArcGIS NRA-tool is based, reduced the number of acronyms used, explained the basics of this framework in order to make clear how this tool works, and we tried to adapt the description of the software to those readers that are not familiar with this framework. 

We would like to emphasize that it is our main aim to present a software tool for research. The research results provided with our manuscript are rather meant as an example and an illustration of the utility and usability of this tool but not as a sophisticated analysis of the national responsibilities countries do have for threatened bird species in Asia and where the main priorities for conservation effort should be set.

We tried to make sure that the manuscript meets PLOS ONE criteria for manuscripts that describe new software for applications following the guidelines mentioned in your decision letter from April 21st of this year.

Utility: From our knowledge, there is no other GIS tool available, which is based on the methods presented by Schmeller et al. in several publications mentioned as references in our manuscript. After publication in PLOS ONE we will open opportunities for active collaboration and further development and overhand the complete source code to the public via github and/or sourceforge securing long term availability and maintenance. The future maintenance, software development and growth will depend on the interest of the scientific community, therefore.

Validation: Until now, there was no option for the automatization and detailed documentation of analyses of National Responsibilities and Conservation Priorities based on the methods described by Schmeller et al. With our ArcGIS-NRA-Tool, these partly difficult and especially time consuming steps of the analyses can be done automatically as demonstrated with the examples described in our manuscript.

Availability: The tool is currently hosted in https://github.com/ and will be made publicly available via https://sourceforge.net/ under the GNU General Public Licence after our manuscript is accepted. The links will be updated in the final version of the manuscript, permissions will be set to “public”, and the source code will be made available. Besides, our tool is already advertised in the web presentation of the EH FP7 SCALES project (www.scales-project.net) and at the web presentation of the SCALES-Taiwan project (http://homepage.ntu.edu.tw/~yplin/Scales-Taiwan.htm). We will amend the links in these web pages too. A long term backup will be made in the archive of the Helmholtz Centre for Environmental Research (UFZ) in Leipzig.

With our revision we are providing a marked-up copy of our manuscript that highlights changes made to the original version (labeled 'Revised Manuscript with Track Changes'), an unmarked version of our revised paper without tracked changes (labeled ‘Manuscript’), four revised and five new figures, one revised and one new table. We have included information from one file of the supplemental material into the main text and dropped this file, revised and renumbered the remaining 4 files and added two new files with results to the supplemental material.

Comments and questions of the reviewers with our answers and respective revisions:

Reviewer #1: 

This manuscript regards the application of a prioritization method together with the national responsibility approach (an approach already used in several scientific works) in a novel GIS tool. This tool is presented with data from Asia. The manuscript is well written but there are a number of major concerns that need to be addressed.

I think that national responsibility and conservation priority approaches should be thoroughly presented. What is national responsibility? How can this be derived? This in addition to the specific explanation given in the method section. There are a number of prioritization methods (including or not responsibility) applied in the literature and a brief overview should be given (see also indications in the discussion section). Data availability and type can influence the applicability of spatial prioritization approaches. This should be introduced as it can be relevant for application of the tool to other regions.

Thank you for the suggestions. We have changed this part in the introduction and extended it by more detailed explanations about national responsibility and approaches for the determination of conservation priorities. However, it is not the aim of our paper to review such approaches but to present a software tool that can be used to perform a certain type of analyses based on methods already described before. This tool should not be used without respective knowledge and experience in this field as it is required for any scientific work.

Methods

There are some aspects that deserve further explanation. Please give more details on what is meant with focal and reference area (I don’t think referring to Schmeller and colleagues suffices). Regarding the additional factor in the prioritization process, I miss how the different scores and classes were decided and assigned. I think that a graphical framework of the National Responsibility Assessment Tool would enable a better understanding of the type of data, steps and requirements needed for its application (maybe integrated in fig. 1).

We have overhauled the whole section and provided new figures and maps explaining the different terms and the process. There is already a graphical framework explaining the workflow (Fig. 2) and the user interface as well as the data needed (S2). We think that a potential user should study all this material provided by us in detail and read the respective references to get familiar with the approaches and the program first. Again, it is not the aim of this paper to explain and discuss approaches for the determination of national responsibilities, we want to introduce a tool. We also separated the more technical parts from the main text and provided them in the supplemental material to make the paper easier to read.

L119-120: I suggest anticipating that this can be used at the global as well as at other scales (regional? national?).

We stated this in an introductory sentence referring to the special case of global analyses and the data needed for this. At the end of the paragraph we were writing:

129 However, analyses can be conducted also at other scales and with other administrative 

130 references.

We moved this sentence to the beginning and inserted a link to Supplement S1 to make this clearer.

L126-127: I do not understand what files authors refer to. The message of this sentence is not clear.

The approach requires the definition of the Reference Area. The borders of this area can be set either by a special file ( e.g. created by a dissolve operation based on a shapefile of the continents of the world) or just by referring in the respective fields in the user interface to either the file used for the Geographical Regions or the Ecological Unit/Region. We suggest to have a glance on Supplement S1 while reading this section because this document shows the respective screenshots of the user interface. We refer to S1 in this paragraph now.

L127: “All layers resp. GIS” is not clear.

We agree that this part of the sentence may be confusing and have tried to write more clearly what we mean. Thank you!

I found really complex to follow the “Focal area and reference area” section. A table could help in reporting what are the different options. 

We now explain this part more in detail with a more detailed description of the method and a set of examples in five new figures.

One additional suggestion is to add the factors used in the applied example(see comment on S1). This would enable also to use as example specific species. It is not clear whether species data can be points, polygons and/or based on grids.

Thank you. However, we clearly wrote:

119 The ArcGIS-NRA-Tool needs four predefined layers with polygons to allow assessments …

It is not possible to use data organized in points or grids, this is not a limitation. It would be just inappropriate to use such type of data with this kind of geographical method.

Even though I do not have a technical background regarding ArcGIS, there is one important point from my side. This tool works in ArcGIS that is a private platform. There are many other programmes such as QGIS and R statistics that are open source and can be accessed without any restrictions. Why didn’t authors decided t produce tools or scripts for open source programmes? I think this would help the application of the approach.

Thank you. We agree with this opinion. For R we don’t see any necessity to provide a special program because an experienced user can do this very easily by itself. ArcGIS is widely used in administrations and scientific institutions and also by organisations like e.g., the IUCN. Therefore we started the development of such a tool with ArcGIS. At this time QGIS was not that widespread and commonly used as it is today. However, we also have already prepared a version for QGIS, which is still in a testing phase but will be released soon too. In the section “Further development” we have added a respective paragraph to the manuscript.

L259-260: It is not clear if one single biogeographic category is assigned to each country.

As mentioned already before, we have tried to explain this part more in detail with a more detailed description of the method and a set of examples in five new figures. The tool provides two different approaches, one is using simple counting of the number of polygons, which are overlapped / touched by the species area and a certain country the other one is performing an intersection among the different layers to calculate the accurate proportions of the overlapping areas.

L260-261: Which scientific reference is used to identify widespread and migratory species?

We didn’t use any reference for that. This sentences simply mean that we didn’t exclude species that are either living in areas bigger than the chosen Reference Area (Asia) = widespread and/or are migratory species, means leaving the Reference Area at least for parts of the year. However, it was not the aim of this study to discuss special aspects of national responsibility assessments related to migration or animal movement, we only want to give an example for the use of our tool. We added a short sentence about this to the manuscript.

Fig. 3: I do not think this map is relevant to understand the proposed process or to show its results! This could be part of one of the Supplementary files.

Thank you for the suggestion. We agree and have dropped this figure from the manuscript.

Overall I think that acronyms do not help the reader.

Thank you! We agree. We only used the abbreviations to keep the text shorter. Because PLOS ONE has no word limits we decided to use abbreviations only if we think it is helpful.

L273: Are the ranges of the number of biogeographic zones used to differentiate between local, regional or widely distributed species fixed? If these are fixed, as I have understood, such ranges should be reported in the method section and removed from the results.

Thank you! We agree and have removed this information from the results but referred to references and a figure in the methods section.

Fig. 4: Not clear what the numbers reported in the figure legend mean. The caption misses this information and what the different colours mean.

Thank you! We have revised and improved this Figure (now Figure 8)

Fig. 5: Not clear what the numbers reported in the figure legend mean. The caption misses this information and what the different colours mean.

Thank you! We have revised and improved this Figure (now Figure 9)

L308-309: Friendly and attractive environment! I think authors can state this once they have users' feedback.

Thank you! We agree and have deleted this statement from the text.

Results for the study case show that in some cases there can be a strong dependency on the national responsibility category (i.e. on species distribution) as many species usually are classified as least concerned. A discussion of this and other possible limitations should be integrated in the discussion.

We don’t understand this sentence. The methodological approach and possible limitations were already addressed in the methodological papers. This is not the aim of our manuscript. There are no such limitations related directly to the tool. Additionally, we have discussed special aspects in Lines 312 to 322 in the revised manuscript.

L323-328: It would be nice to have some results on this overlap in the supplementary material.

Thank you for the suggestion. This might be a misunderstanding. We didn’t do any special analyses in this regard. We only linked our results to information provided by the papers cited. We revised this paragraph to make this more clear. However, now we provide still more data from the output files produced by our tool (Supplement S5) and some statistics, which will be presented in an additional table in the manuscript (Table 2), and supported by Supplement S6. This material is used for a more detailed interpretation, which we have added to this section.

I suggest to highlight the possibility or not of adding some indicators of conservation priority. For example, current and future acting forces can help to set priorities also in terms of conservation and management activities. There are a number of works that use these factors to identify priorities (e.g. examples from different continents: Zhang et al. 2014; Campagnaro et al. 2018; Carvalho et al. 2020). These works can be used to make additional comments on the need of spatial information and compare different proposed tools. These works can also help in highlighting possible integrations to the spatial tool that authors are presenting.

Thank you for your suggestions. However, although it is not the aim of the paper to review different other methods and tools, we have integrated a paragraph giving a short overview in the introduction, provided some ideas for integration of appropriate spatial conservation prioritization approaches in the section “Further development” and referred to other software tools, types of analyses and methodological approaches, which could be used in sophisticated combined approaches in the discussion section.

S1: I think that some examples can be reported in the main text (see comments on the method section). For example, “Reference Area: Asia border” and “Focal Area: Asian countries”. This would benefit comprehension of the national responsibility tool.

and

S1 L18-20: I suggest reporting these possible issues in the main text.

We are grateful for these suggestions and have moved the whole content of Supplement S1 to the part “Technical requirements” in the section “Description of the National Responsibility Assessment Tool” of the main text. Additionally, as mentioned above, we have already added some new paragraphs to explain the approach a little bit more in detail with an other example in the “Methods” section.

S2 L12: Small typo: “CNTRY_NAME”.

This is not a typo, it is the name of the field for the Focal Region you can see in right panel of the user interface in Figure S2.1. Shapefiles use dBase format for attribute tables. dBase field names are limited to the string length of 10 characters only. Because we have moved the whole content of S1 to the main text S2 is now S1.

S3: Are the reported examples of possible cases already presented in other publications? I find this supplementary extremely useful to better understand possible cases.

Thank you! No, these examples are purely theoretical and have not been published so far. We only used them to better illustrate the difference between both approaches addressed in it. We feel that this artificial construct is more helpful to understand the problems we have to deal with than an example with real maps and species distributions where the reader can be distracted by shape and size of a certain polygon. However, we felt it is better to keep it separated from the main text. Because we have moved the whole content of S1 to the main text S3 is now S2.

Reviewer #2: 

Thank you for the opportunity to review this manuscript. This study develops and demonstrates a GIS-based tool, which can be used to produce National Responsibilities and Conservation Priorities for multiple species using freely available data. For me, this study demonstrates a great tool, which is relatively easy to use and has a pretty good flexibility to be applied on different scales and locations. Furthermore, using freely available species distribution and ecoregion data and setting a standardized assessment for countries are the strengths of this tool. In my opinion, this can certainly contribute to setting conservation prioritizations for policy makers.

Although I know the focus of this study is about the tool itself, I have a question about the method behind this tool, which is the issue about wide-ranging nomadic species and migratory species. Could this method cause an issue like “tragedy of the commons” for wide and evenly distributed species? For example, would wide-ranging nomadic species be considered as basic in NR throughout all countries it lives? How would you address this issue in the method?

Thank you for raising this question. Indeed, a wide ranging or migratory species with a distribution area covering many countries and eco-regions would be probably considered rather as basic throughout all countries it lives. That means that no country would have a special National Responsibility. However, in this case the IUCN Red List status would have more weight and the burden should be shared between all countries. It doesn’t mean that this species must not be protected, it only means that not one or some more countries would have a higher responsibility because the species mainly occurs in it or them respectively. The scores of National Responsibility will be correlated with the size of the countries in this case. The country with the largest area would also have the highest responsibility.

Especially for migratory species, detailed analyses could be made focusing either only on the breeding range, areas which are either important during the time of migration or in winter. There are a lot of options for such analyses. However, this is not a question of the tool or special options in it, it is rather a question of selection of species, geographical focus, and stratification in the data. Hence it is in the responsibility of the researcher and the design of the study.

To address and answer such questions we have integrated already options to change the parameters for the analyses (Supplement S1, Fig. S1.2b) and the two approaches. However, it takes more special investigations with standardised theoretical and real data as well to find out more about such effects and possible causes of bias. This is planned for the future and also one reason why we want to provide this tool to the community and other researchers to address such problems.

In Runge et al. (2015), they suggest that it might be more suitable to use the minimum range nomads occupy across multiple years because of their fluctuated distribution. The minimum range might contain some climate refugial sites or high quality habitats for those wide-ranging species.

Runge et al. (2015) Geographic range size and extinction risk assessment in nomadic species. Conservation Biology

Thank you. This is an interesting aspect. However, the tool presented is not limited regarding this. It depends on research question, knowledge of the researcher, aim of the study etc., which polygons will be used for the analysis. Our tool only helps to automatize some steps within this work flow it does not answer questions or make decisions the researcher should do.

For migratory species, I’m curious about how you deal with them in the study. Would you mind to provide some information about this in the manuscript? since this study also includes several migratory species. Do you use their breeding ranges, non-breeding ranges, or passage ranges, or all combined, or all but separated (which means one migratory bird might be counted as NR and CP in countries where it breeds, winters, and passes by). I suppose the last method, which is using all ranges separately, make more sense because migratory populations rely on all habitats for different life stages. However, some migratory birds in Asia might have their non-breeding ranges or passage ranges extend to Oceania (e.g., Australia and New Zealand)

This may influence your results (Fig. 4, and 5) because the proportion of migratory birds among all avian communities can be very high in the north hemisphere. See Somveille et al. (2013) Mapping Global Diversity Patterns for Migratory Birds. PLOS ONE.

Thank you for raising this question. Indeed, migratory species need a special treatment and more sophisticated analyses and interpretation of results in such studies. Depending on the question a researcher has, it is possible to use separate breeding, wintering and passing by ranges with the tool, if the researcher separates these data in the input files. However, our example was only meant as an illustration of the opportunities. It was not our aim to do a detailed analysis in this case. This is planned for the future. We only mentioned migratory and widespread species because we didn’t want to exclude such species from the analysis. A complete assessment would require the detailed discussion of each species and, therefore, go far beyond the aim of our manuscript.

Minor comments as follows

line 32: to prioritize conservation what? to prioritize conservation resources? funds? effort? Actions?

Thank you! We have specified this in the respective sentence.

line 54: could you provide one or a few sentences in the introduction to define or explain “national responsibility”? If I didn’t read any previous studies about NR, I would only realize what it really is until the Methods…

We have added a sentence to explain this concept. Thank you!

line 184: While these two BUs may be useful for terrestrial species, incorporating “Freshwater Ecoregions of the World” might further enlarge the applicability of this tool. The distribution of freshwater biodiversity maybe more relevant to the shape and size and boundary of watersheds/river basins than local climate or terrestrial vegetation. The freshwater ecoregion data is also freely available online. See Abell et al. (2008)

Freshwater ecoregions of the world: A new map of biogeographic units for freshwater biodiversity conservation. BioScience.

Thank you for mentioning this aspect. Indeed, the tool and this approach are not limited to terrestrial ecosystems. We have added a paragraph explaining this and referring to resources for freshwater and marine ecoregions.

line 312: Which part of your results supports this statement about “sensitive”? Is there any quantified sensitivity? such as sensitivity analysis or a plot about the correlation between the range size of FS and biophysical regions and NR? or simply based on the description.

We didn’t do any specific sensitivity analysis. However, we got already some experience in former analyses and one also can derive such effects simply from the type of analysis and distribution of polygons. However, as already mentioned before in this letter and also mentioned at the end of the discussion in our manuscript, one aim to develop and release this tool was to support such type of studies. We added additional sentences at the end of the “Discussion” section in the manuscript.

line 528: “probability” or “responsibility”?

Thank you! We have revised to “responsibility”

Additional information

The program will be described in github.com and sourceforge.org with the following text:

ArcGIS-NRA-Tool

ArcGIS Tool for the Assessment of National Responsibilities for Endangered Species and Habitats

The aim of the source code is to develop a National Responsibility Assessment Tool that can be used as plug-in for ArcGIS (https://github.com/popecologist/ArcGIS-NRA-Tool and http://popecologist.github.io respectively or https://sourceforge.net/projects/arcgis-nra-tool/).

There are two binary versions of the add-in: SpeciesTool.NET.20140619.esriAddIn - Version from 2014-06-14 compiled with Visual Studio 12, Microsoft .NET 3.4, and the matching (older) SDKs ArcGISRuntime and ArcObjects from ESRI. SpeciesTool.NET.20200228.esriAddIn - Version from 2020-02-28 compiled with Visual Studio 17, Microsoft .NET 4.6.1, and the and the matching (newer) SDKs ArcGISRuntime 100.2.0 and ArcObjects for ArcGIS 10.6.1 from ESRI.

Copyright (c) 2020 Yu-Pin Lin / 林裕彬 Department of Bioenvironmental Systems Engineering National Taiwan University No. 1, Sec. 4, Roosevelt Road Taipei 10617 Taiwan R.O.C. Office Phone: 886-2-33663467; Fax: 886-2-2368-6980 E-mail: yplin@ntu.edu.tw
http://homepage.ntu.edu.tw/~yplin/Scales-Taiwan.htm

Credits: Academia Sinica (programming) Reinhard A. Klenke (revising, updating)

The development of the ArcGIS-NRA-Tool was mainly funded by Ministry of Science and Technology of Taiwan (former National Science Council of Taiwan, code NSC101-2923-I-002-001-MY2), and a contribution from the EU FP7 project SCALES: Securing the Conservation of biodiversity across Administrative Levels and spatial, temporal, and Ecological Scales, under the European Union’s Framework Program 7 (Code: 226852 FP7-ENVIRONMENT ENV.2008.2.1.4.4., www. scales-project.net.

This program is free software: you can redistribute it and/or modify it under the terms of the GNU General Public License as published by the Free Software Foundation, version 3. This program is distributed in the hope that it will be useful, but WITHOUT ANY WARRANTY; without even the implied warranty of MERCHANTABILITY or FITNESS FOR A PARTICULAR PURPOSE. See the GNU General Public License for more details.

You should have received a copy of the GNU General Public License along with this program. If not, see http://www.gnu.org/licenses/. Please cite and refer to the tool in any report, scientific paper or other type of publication either in print or electronically with the reference mentioned below.

Unfortunately we still need proprietary libraries from third parties such as ESRI (e.g. ArcGIS SDK, ArcObjects SDK) and from Microsoft to be linked again this source code. The need for these publicly available but proprietary libraries is a drawback. Therefore, we suggest to anyone who thinks of doing substantial further work on the program to give highest priority to tasks changing the program in a way that it can do the same job without the proprietary libraries.

Installation of the ArcGIS-add-in: There might be a "No GUI components found in this Add-In. Add-In version does not match." error during installation. This error only occurs when a user tries to install the add-in using the in the Customize dialog. When the add-in is double-clicked from Windows Explorer, the add-in is installed successfully. This is a defect (NIM095435 http://support.esri.com/bugs/nimbus/TklNMDk1NDM1. A potential workaround is to either: the Add-In in Windows Explorer to install OR add folder through in Add-In Manager (see more at: https://community.esri.com/thread/162324).

References: Lin Y-P, Schmeller D S, Ding T S, Wang Y Ch, Lien W-Y, Henle K, Klenke R A (2020): A GIS-based policy support tool to determine national responsibilities and priorities for biodiversity conservation. PLOS ONE. doi: xxxxxx

Each separate file of the C# code is starting with the following lines referring to the GNU General Public License and to the Article in PLOS ONE:

/*

 * This file is part of the ArcGIS National Responsibility Assessment Tool

 * source code (https://github.com/popecologist/ArcGIS-NRA Tool or http://xxx.github.io).

 *

 * Copyright (c) 2020 Yu-Pin Lin / 林裕彬

 * Department of Bioenvironmental Systems Engineering

 * National Taiwan University

 * No. 1, Sec. 4, Roosevelt Road

 * Taipei

 * 10617 Taiwan

 * R.O.C.

 * Office Phone: 886-2-33663467; Fax: 886-2-2368-6980

 * E-mail: yplin@ntu.edu.tw

 * http://homepage.ntu.edu.tw/~yplin/Scales-Taiwan.htm

 * 

 * Credits:

 * Academia Sinica (programming)

 * Reinhard A. Klenke (revising, updating)

 *

 * The development of the ArcGIS-NRA-Tool was mainly funded

 * by Minister of Science and Technology of Taiwan

 * (National Science Council of Taiwan) (NSC101-2923-I-002-001-MY2),

 * and a contribution from the EU FP7 project

 * SCALES: Securing the Conservation of biodiversity across

 * Administrative Levels and spatial, temporal, and Ecological Scales,

 * under the European Union’s Framework Program 7

 * (grant Code: 226852 FP7-ENVIRONMENT ENV.2008.2.1.4.4.;

 * www. scales-project.net.

 *

 * This program is free software: you can redistribute it and/or modify

 * it under the terms of the GNU General Public License as published by

 * the Free Software Foundation, version 3.

 *

 * This program is distributed in the hope that it will be useful, but

 * WITHOUT ANY WARRANTY; without even the implied warranty of

 * MERCHANTABILITY or FITNESS FOR A PARTICULAR PURPOSE. See the GNU

 * General Public License for more details.

 *

 * You should have received a copy of the GNU General Public License

 * along with this program. If not, see <http://www.gnu.org/licenses/>.

 *

 * Please cite and refer to the tool in any report, scientific paper or

 * other type of publication either in print or electronically with the

 * reference mentioned below.

 *

 * References:

 * Lin Y-P, Schmeller D S, Ding T S, Wang Y Ch, Lien W-Y, Henle K,

 * Klenke R A (2020): A GIS-based policy support tool to determine national

 * responsibilities and priorities for biodiversity conservation.

 * PLOS ONE. doi: xxxxxx

 */

---

## [Decision Letter · Decision Letter 1]

14 Oct 2020

PONE-D-20-05865R1

A GIS-based policy support tool to determine national responsibilities and priorities for biodiversity conservation

PLOS ONE

Dear Dr. Klenke,

Thank you for submitting your manuscript to PLOS ONE. After careful consideration, we feel that it has merit but does not fully meet PLOS ONE’s publication criteria as it currently stands. Therefore, we invite you to submit a revised version of the manuscript that addresses the points raised during the review process.

The previous reviewers have reviewed the new version of the manuscript, and both agree that the paper is much improved, and that the authors have addressed the previous comments seriously and effectively. Nevertheless, they also suggest some minor yet important revisions to your manuscript that should be addressed. Therefore, I invite you to respond to the reviewer' comments and revise your manuscript.

Overall, reviewers agree that the manuscript needs a deep review for language/grammar. I would highly encourage you to seek editorial help or have a native English speaker review of the manuscript before next submission.

We look forward to receiving your revised manuscript.

Kind regards,

Pedro Abellán

Academic Editor

PLOS ONE

Reviewers' comments:

Reviewer's Responses to Questions

**Comments to the Author**

1. If the authors have adequately addressed your comments raised in a previous round of review and you feel that this manuscript is now acceptable for publication, you may indicate that here to bypass the “Comments to the Author” section, enter your conflict of interest statement in the “Confidential to Editor” section, and submit your "Accept" recommendation.

Reviewer #1: (No Response)

Reviewer #2: (No Response)

2. Is the manuscript technically sound, and do the data support the conclusions?

Reviewer #1: Yes

Reviewer #2: Yes

3. Has the statistical analysis been performed appropriately and rigorously? 

Reviewer #1: Yes

Reviewer #2: N/A

4. Have the authors made all data underlying the findings in their manuscript fully available?

Reviewer #1: Yes

Reviewer #2: Yes

5. Is the manuscript presented in an intelligible fashion and written in standard English?

Reviewer #1: Yes

Reviewer #2: No

6. Review Comments to the Author

Reviewer #1: I thank the authors for considering all my comments and following most of them. Authors made some important changes based on suggestions of both reviewers and I think the manuscript has greatly improved compared to the former version. I really hope this tool will find large use in the biodiversity conservation world.

However, I still have some minor comments. I final language revision is needed.

i. I think you should avoid mentioning “Schmeller and collegues” in the abstract.

ii. L120-122, L500-501: What about the four species of shrews? (e.g. L141-143).

iii. L503: Repetition: “provides users an user interface”

iv. L508: It is a strange sentence and a bit redundant (“the bunch of methods mentioned in the introduction already before”).

v. L530-531: Which group? Do you mean medium/small size countries?

vi. L560: I think that it should be “..earlier by Schmeller et al. [1]”. There are other cases in the text.

Good luck and best regards

Reviewer #2: Thank you for this opportunity to review this work again. Generally, this manuscript is much improved and being revised according to suggestions and comments from reviewers. Specifically, now the authors clearly explained what national responsibility is and clarified the differences between this approach and other prioritization methods for biodiversity conservation. I generally satisfied with this revised version especially with the science part (which is more important for me). Nevertheless, I highly recommend authors be more careful on their writing and do a thorough check because there are many typos, redundant words, and confusing sentences throughout the manuscript (maybe more than the first version). Although I personally do not think these typos would heavily influence my decision for this manuscript, in some cases these small mistakes could severely reduce the readability of a great work.

While I only listed a few things I saw here, a more thorough check is highly recommended.

Lines 54-56 and 57-61: same or different paragraphs?

Line 78-89: While I really appreciate the comprehensiveness of this long list about "other methods", it is a bit too long and thus interrupts the logic flow of the Introduction...

Could you make this list a bit shorter (maybe group some of these into a few groups?! e.g., species-focusing, habitat-focusing, socioeconomic-focusing, mixed approaches,...etc)? or add one sentence at the end of this paragraph to guide the focus of readers back to national responsibilities (something like line 111-113)?

Line 112: “like” “e.g.” redundant. Please pick one

Line 112: “,” after Zonation

Line 113: they do not XXX for the estimation… a verb is lacking here

Line 119-120: Could you rewrite this sentence? It is a bit strange...something like “…for a large set of species for which data are available such as the species range data in the IUCN Red List database”

Ling 127: suggest deleting “However”

Line 134: “like” “e.g.” redundant. Please pick one

Line 136-137: “only” redundant

Line 141: suggest deleting “However”

Line 141: suggest using past tense because you have already done it

Line 163-164: “also””too” redundant

Line 191-195: suggest dividing this long sentence into two short sentences to improve the readability

Line 210-212: I am not sure what does this sentence mean…

Line 212: For Sorex minutus”,”

Line 213: “the” highest responsibility

Line 213: suggest replacing “the” with “this”

Line 225: suggest deleting “of course”

Line 225: suggest deleting “also”

Line 228: suggest deleting “area”

Line 235: suggest changing to something like

“…or in the same projection for analyses on a regional scale to provide...”

Line 239: suggest deleting “animal and plant”

Line 243: suggest changing “…on data freely available from different sources” to something like

“…or on freely available data.”

Line 245: suggest changing “ in shapefile format from (<- a typo here)” to “as shapefiles”

Line 245-246: suggest changing to “The distribution of each species or habitat category should be saved individually for the analyses”

Line 248: suggest changing to “files downloaded from the IUCN...”

Line 260: suggest deleting “of this type”

Line 284: suggest changing “in the form of” to “as”

Line 298: suggest “we strongly suggest using the GEnZ level…”

Line 302-307: suggest turning this whole paragraph into one or two sentences at the end of previous paragraph

Line 328: suggest deleting “still”

Line 329: suggest deleting “the” right before Biogeographic Units

Line 338: suggest deleting “those”

Line 348: suggest deleting the first “the”

Line 361: suggest adding “of” right behind “layers”

Line 364: suggest using “e.g.,” within parentheses and “for example” outside parentheses. Nevertheless, no matter which one you prefer, be consistent

Line 379-380: This sentence is identical with line 369-371 but with different references? One could be deleted

Line 403: suggest deleting “still”

Line 405: “such as” “e.g.” redundant. Please pick one

Line 439-440: “also” “as well” redundant. Please pick one

Line 442: Why are there two China here…

Line 443: Vietnam”,”

Line 458: suggest deleting “more”

Line 458-467: I'm not sure what is the importance of this detailed quantified comparison described in this paragraph and table 2 if there is NO DISCUSSION following up?

The different results from PA-A and PC-A are expected because they use different information...

Suggest moving most part of this paragraph and table 2 to supplementary files. A few sentences describing the overall differences/patterns between these two approaches (like what you did for “Conservation priority”) and Figs 8 and 9 should be enough.

Alternatively (you don't need to do this, just a suggestion), an analysis on the relationship between the change of ranking versus the environmental variability (heterogeneity) across countries might be more interesting...because the authors DID mention environmental variability could be one of the reasons causing the differences between these two approaches.

Line 507: “like” “e.g.” redundant. Please pick one

Line 548: suggest changing “focus” to “allocate”

Line 549-550: “…also including guiding capacity building” not sure what does this mean

Line 553: suggest changing “…for which we have distribution data available” to “…for which distribution data are available”

7. PLOS authors have the option to publish the peer review history of their article (what does this mean?). If published, this will include your full peer review and any attached files.

Reviewer #1: No

Reviewer #2: No

---

## [Author Response · Author response to Decision Letter 1]

14 Nov 2020

Comments and questions of the reviewers with our answers and respective revisions

PONE-D-20-05865R1 

A GIS-based policy support tool to determine national responsibilities and priorities for biodiversity conservation 

PLOS ONE 

...

Overall, reviewers agree that the manuscript needs a deep review for language/grammar. I would highly encourage you to seek editorial help or have a native English speaker review of the manuscript before next submission.

Thank you. We asked a native speaker from our team for a check of language and grammar and tried to improve the text based on her suggestions.

Reviewer #1: I thank the authors for considering all my comments and following most of them. Authors made some important changes based on suggestions of both reviewers and I think the manuscript has greatly improved compared to the former version. I really hope this tool will find large use in the biodiversity conservation world. 

However, I still have some minor comments. I final language revision is needed. 

We want to thank reviewer #1 for the very helpful suggestions and tried to carefully consider all of them in our revision. The manuscript was also checked by a native speaker and revised substantially.

i. I think you should avoid mentioning “Schmeller and colleagues” in the abstract.

We changed the introduction.

ii. L120-122, L500-501: What about the four species of shrews? (e.g. L141-143). 

Thank you, we changed it to: “To demonstrate the use of our tool and to illustrate potential pitfalls, we illustrate our approach using four shrew species with different distribution areas (Fig 1) and 258 Asian bird species”

iii. L503: Repetition: “provides users an user interface” 

We changed it to: “provides users with an interface” 

iv. L508: It is a strange sentence and a bit redundant (“the bunch of methods mentioned in the introduction already before”). 

We deleted this part.

v. L530-531: Which group? Do you mean medium/small size countries? 

We changed it to: “showed that countries with at least “medium” responsibility for a high number of species fall in conservation priority Class 2.”

vi. L560: I think that it should be “..earlier by Schmeller et al. [1]”. There are other cases in the text. 

We revised all cases and either changed the wording and removed by/according or we mentioned the authors.

Reviewer #2: Thank you for this opportunity to review this work again. Generally, this manuscript is much improved and being revised according to suggestions and comments from reviewers. Specifically, now the authors clearly explained what national responsibility is and clarified the differences between this approach and other prioritization methods for biodiversity conservation. I generally satisfied with this revised version especially with the science part (which is more important for me). Nevertheless, I highly recommend authors be more careful on their writing and do a thorough check because there are many typos, redundant words, and confusing sentences throughout the manuscript (maybe more than the first version). Although I personally do not think these typos would heavily influence my decision for this manuscript, in some cases these small mistakes could severely reduce the readability of a great work. 

While I only listed a few things I saw here, a more thorough check is highly recommended. 

We want to thank reviewer #2 for the numerous and very helpful suggestions and the high effort in proof reading of our text. We tried to carefully consider almost all of them in our revision. We also beg your pardon for submitting a manuscript still containing so many language issues. To improve grammar, orthography, and style, the manuscript has been checked by a native speaker and revised substantially by the authors.

Lines 54-56 and 57-61: same or different paragraphs? 

We merged it to one paragraph.

Line 78-89: While I really appreciate the comprehensiveness of this long list about "other methods", it is a bit too long and thus interrupts the logic flow of the Introduction... 

Could you make this list a bit shorter (maybe group some of these into a few groups?! e.g., species-focusing, habitat-focusing, socioeconomic-focusing, mixed approaches,...etc)? or add one sentence at the end of this paragraph to guide the focus of readers back to national responsibilities (something like line 111-113)? 

Thank you, we compiled the list in a table to separate it from the text.

Line 112: “like” “e.g.” redundant. Please pick one 

We deleted “e.g.”

Line 112: “,” after Zonation 

We inserted a comma.

Line 113: they do not XXX for the estimation… a verb is lacking here 

Thank you, we changed the sentence.

Line 119-120: Could you rewrite this sentence? It is a bit strange...something like “…for a large set of species for which data are available such as the species range data in the IUCN Red List database” 

Thank you, we changed it to: “such as the Spatial Data & Mapping Resources and the Red List database from IUCN [41, 42]”

Ling 127: suggest deleting “However” 

We deleted “However”

Line 134: “like” “e.g.” redundant. Please pick one 

We deleted “e.g.”

Line 136-137: “only” redundant 

We deleted “only”

Line 141: suggest deleting “However” 

We deleted “However”

Line 141: suggest using past tense because you have already done it 

We changed the tense.

Line 163-164: “also””too” redundant 

We deleted “too”

Line 191-195: suggest dividing this long sentence into two short sentences to improve the readability 

We split the sentence in two shorter ones.

Line 210-212: I am not sure what does this sentence mean… 

Thank you. We rewrote the sentence: “Sorex alpinus shows preferences for alpine habitats that are less common in the focal area (France) but comprise large regions of Switzerland and Austria. “

Line 212: For Sorex minutus”,” 

We inserted a comma.

Line 213: “the” highest responsibility 

We revised to: “For Sorex minutus, Russia scores very high on the responsibility metric ...”

Line 213: suggest replacing “the” with “this” 

We replaced “the” with “this”.

Line 225: suggest deleting “of course” 

We have replaced “of course” with “such”

Line 225: suggest deleting “also” 

Thank you. However, we think, deleting “also” would change the meaning of this sentence and left it as it is.

Line 228: suggest deleting “area” 

We deleted “area”.

Line 235: suggest changing to something like 

“…or in the same projection for analyses on a regional scale to provide...” 

Thank you. We changed it to: “…or in the same equal area projection for analyses on a smaller scale to provide...” 

Line 239: suggest deleting “animal and plant” 

We deleted “animal and plant”

Line 243: suggest changing “…on data freely available from different sources” to something like 

“…or on freely available data.” 

Thank you. We followed this suggestion.

Line 245: suggest changing “ in shapefile format from (<- a typo here)” to “as shapefiles” 

We changed the sentence.

Line 245-246: suggest changing to “The distribution of each species or habitat category should be saved individually for the analyses” 

Thank you. We changed the two sentences referring to this issue to one, which better explains what we mean, hopefully.: “… Examples of sources include the webservices of IUCN (https://www.iucnredlist.org/) and Map of Life (https://www.mol.org) [52]. The distribution of each species or habitat category must be available as a separate layer for use in our tool.”

Line 248: suggest changing to “files downloaded from the IUCN...” 

We followed your suggestion and changed it to: “Files downloaded from the IUCN webserver... “

Line 260: suggest deleting “of this type” 

We deleted “of this type”.

Line 284: suggest changing “in the form of” to “as” 

We changed “in the form of shapefiles” to: “as separate layers”, which we think is more appropriate.

Line 298: suggest “we strongly suggest using the GEnZ level…” 

We followed your suggestion.

Line 302-307: suggest turning this whole paragraph into one or two sentences at the end of previous paragraph 

Thank you. However, we think it would be better to break this down into two paragraphs as it will make it more clear to the reader that the tool has no limitations and that sources are available for ecosystems other than just terrestrial.

Line 328: suggest deleting “still” 

We deleted “still”.

Line 329: suggest deleting “the” right before Biogeographic Units 

We deleted “the” right before Biogeographic Units.

Line 338: suggest deleting “those” 

We deleted “those”.

Line 348: suggest deleting the first “the” 

We deleted the first “the”.

Line 361: suggest adding “of” right behind “layers” 

We changed the whole paragraph starting with: “Files to be used with the NRA-Tool must be named according to the following convention: ...”

Line 364: suggest using “e.g.,” within parentheses and “for example” outside parentheses. Nevertheless, no matter which one you prefer, be consistent 

Thank you. We followed your suggestions and changed “for example” to ““e.g.,” in parentheses.

Line 379-380: This sentence is identical with line 369-371 but with different references? One could be deleted 

We deleted the sentence in lines 369-371.

Line 403: suggest deleting “still” 

We deleted “still”.

Line 405: “such as” “e.g.” redundant. Please pick one 

We deleted “such”.

Line 439-440: “also” “as well” redundant. Please pick one 

We deleted “also”

Line 442: Why are there two China here… 

t’s a typo. We deleted one China.

Line 443: Vietnam”,” 

We inserted a comma after Vietnam.

Line 458: suggest deleting “more” 

Thank you. We deleted “more”

Line 458-467: I'm not sure what is the importance of this detailed quantified comparison described in this paragraph and table 2 if there is NO DISCUSSION following up? 

The different results from PA-A and PC-A are expected because they use different information... 

Suggest moving most part of this paragraph and table 2 to supplementary files. A few sentences describing the overall differences/patterns between these two approaches (like what you did for “Conservation priority”) and Figs 8 and 9 should be enough. 

Alternatively (you don't need to do this, just a suggestion), an analysis on the relationship between the change of ranking versus the environmental variability (heterogeneity) across countries might be more interesting...because the authors DID mention environmental variability could be one of the reasons causing the differences between these two approaches. 

Thank you. We have inserted a new paragraph addressing the results and issues raised by the reviewer respectively: “For more in-depth analyses and interpretation, the tool provides the results not only in map form but also in tables containing all input information, calculated areas, and resulting scores (Table 3, S5 and S6). These outputs can provide insight into the discrepancies between the results of the two methodical approaches offered by our tool. These two approaches should be compared and evaluated in further methodological studies focusing on statistical effects and the functional relation between environmental variability across countries and value distributions. 

Our own analysis shows that the more advanced PA-A provides more differentiated results than the PC-A because it takes area of overlap into account rather than simply using the number of overlapping polygons. This leads to differences in weighting, especially when large countries have relatively little overlap with focal and reference areas. The size of a country also plays a role when considering the number of species for which basic responsibility must be assumed. The larger the territory of a country, the greater the probability that it will include significant portions of the range of a species in the corresponding reference area—a species for which the country then bears basic responsibility. ”

Line 507: “like” “e.g.” redundant. Please pick one 

We deleted “e.g.”

Line 548: suggest changing “focus” to “allocate” 

We changed the sentence, moved it to the end of the section and wrote: “With this information, policy and decision makers will be better equipped to assess when and how policy should be adapted, reinforced, or developed to fulfill the Convention on Biological Diversity’s Aichi targets.”

Line 549-550: “…also including guiding capacity building” not sure what does this mean 

Thank you. Please see our reply to your comment to Line 548 above.

Line 553: suggest changing “…for which we have distribution data available” to “…for which distribution data are available” 

We followed this suggestion.

---

## [Editor Report · Decision Letter 2]

17 Nov 2020

A GIS-based policy support tool to determine national responsibilities and priorities for biodiversity conservation

PONE-D-20-05865R2

Dear Dr. Klenke,

We’re pleased to inform you that your manuscript has been judged scientifically suitable for publication and will be formally accepted for publication once it meets all outstanding technical requirements.

Kind regards,

Pedro Abellán

Academic Editor

PLOS ONE
---

## [Editor Report · Acceptance letter]

24 Nov 2020

PONE-D-20-05865R2 

A GIS-based policy support tool to determine national responsibilities and priorities for biodiversity conservation 

Dear Dr. Klenke:

I'm pleased to inform you that your manuscript has been deemed suitable for publication in PLOS ONE. Congratulations! Your manuscript is now with our production department. 

Kind regards, 

on behalf of

Dr. Pedro Abellán 

Academic Editor

PLOS ONE